# Learning to Incentivize Improvements from Strategic Agents

**Yatong Chen**                                                                                  *ychen592@ucsc.edu*
*Computer Science and Engineering*
*University of California, Santa Cruz*

**Jialu Wang**                                                                                   *faldict@ucsc.edu*
*Computer Science and Engineering*
*University of California, Santa Cruz*

**Yang Liu**                                                                                     *yangliu@ucsc.edu*
*Computer Science and Engineering*
*University of California, Santa Cruz*

**Reviewed on OpenReview:** *https://openreview.net/forum?id=W98AEKQ38Y*

## Abstract

Machine learning systems are often used in settings where individuals adapt their features to obtain a desired outcome. In such settings, strategic behavior leads to a sharp loss in model performance in deployment. In this work, we aim to address this problem by learning classifiers that encourage decision subjects to change their features in a way that leads to improvement in both predicted *and* true outcome. We frame the dynamics of prediction and adaptation as a two-stage game, and characterize optimal strategies for the model designer and its decision subjects. In benchmarks on simulated and real-world datasets, we find that classifiers trained using our method maintain the accuracy of existing approaches while inducing higher levels of improvement and less manipulation.

## 1 Introduction

Individuals subject to a classifier's predictions may act strategically to influence their predictions. Such behavior, often referred to as *strategic manipulation* (Hardt et al., 2016a), may lead to sharp deterioration in classification performance. However, not all strategic behavior is detrimental: in many applications, model designers stand to benefit from strategic adaptation if they deploy a classifier that incentivizes decision subjects to perform adaptations that improve their true outcome (Haghtalab et al., 2020; Shavit et al., 2020). For example:

- **Lending**: In lending, a classifier predicts a loan applicant's ability to repay their loan. If the classifier is designed so as to incentivize the applicants to improve their income, it will also improve the likelihood of repayment.

- **Content Moderation**: In online shopping, a recommender system suggests products to customers based on their relevance. Ideally, the algorithm should incentivize the product sellers to publish accurate product descriptions by aligning this with improved recommendation rankings.

- **Course design**: an instructor designs schoolwork to incentivize students to invest their efforts on studying rather than cheating on an exam (Kleinberg & Raghavan, 2020).

- **Car insurance determination**: an auto insurer tries to predict drivers' expected accident costs, and by designing a determination criterion, encourages safe driving behavior (Haghtalab et al., 2020; Shavit et al., 2020).

In this work, we study the following mechanism design problem: a *model designer* needs to train a classifier that will make predictions over *decision subjects* who will alter their features to obtain a specific prediction. Our goal is to learn a classifier that is accurate and that incentivizes decision subjects to adapt their features in a way that improves both their predicted *and* true outcomes. Our main contributions are as follows:

1. We introduce a new approach to handle strategic adaptation in machine learning, based on a new concept we call the *constructive adaptation risk*, which trains classifiers that incentivize decision subjects to adapt their features in ways that improve true outcomes. Under the assumption of a feature taxonomy that distinguishes improvable features (features that, if changed, lead to changes in the true qualification) from non-causal features (which do not lead to changes in the true qualification), we provide formal evidence that this risk captures both the strategic and constructive dimensions of decision subjects' behavior.

2. We characterize the dynamics of strategic decision subjects and the model designer in a classification setting using a two-player sequential game. We begin by generalizing cost functions used in previous works on strategic classification to the *Mahalanobis* distance, which provides a way to capture correlations between changes in different features. Under this generalization, we derive closed-form expressions for the decision subjects' optimal strategies (Theorem 1). These expressions (Section 3.3) reveal insights about decision subjects' behavior when the model designer uses non-causal features (features that do not affect the true outcome) as predictors.

3. We formulate the problem of training such a desired classifier as a risk minimization problem. We evaluate our method on simulated and real-world datasets to demonstrate how it can be used to incentivize improvement or discourage adversarial manipulation. Our empirical results show that our method outperforms existing approaches, even when some feature types are misspecified. In addition, we provide a potential way to extend our main result into a non-linear setting using LIME (Ribeiro et al., 2016).

The details for reproducing our experimental results can be found at
`https://github.com/UCSC-REAL/ConstructiveAdaptation`.

## 1.1 Related work

Our paper builds on the strategic classification literature in machine learning (Hardt et al., 2016a; Cai et al., 2015; Ben-Porat & Tennenholtz, 2017; Chen et al., 2018; Dong et al., 2018; Dekel et al., 2010; Chen et al., 2020; Tsirtsis et al., 2019). We study the interactions between a model designer and decision subjects using a sequential two-player Stackelberg game (see e.g., Hardt et al., 2016a; Brückner & Scheffer, 2011; Balcan et al., 2015; Dong et al., 2018, for similar formulations). Departing from previous work, which aims to suppress *all* adaptations, we consider a setting in which strategic adaptation can consist of manipulation as well as improvement. Our broader goal of designing a classifier that encourages improvement is characteristic of recent work in this area (see e.g., Kleinberg & Raghavan, 2020; Haghtalab et al., 2020; Shavit et al., 2020; Rosenfeld et al., 2020). Specifically, Haghtalab et al. (2020) study how to design an evaluation mechanism that incentivizes individuals to improve a desired quality. However, the success of their method requires explicit assumptions on the linear mapping of features to true qualifications, as well as a projection matrix $P$ that maps the observed features back to the full features. Their setting also does not account for correlations between different features. Another recent work by Shavit et al. (2020) also focuses on finding a decision rule that maximizes decision subjects' true qualifications. Their setting is similar to ours, but they focus on how decision makers can perform causal interventions through the deployment of different decision rules, rather than designing a classifier relying only on observational data. Moreover, they assume that decision subjects take actions in some *action space* that maps linearly to features in *feature space*; this also does not capture correlations between features.

This paper also broadly relates to work on *recourse* (Ustun et al., 2019; Venkatasubramanian & Alfano, 2020; Karimi et al., 2020a; Gupta et al., 2019; Karimi et al., 2020b; von Kügelgen et al., 2020). Formally speaking, recourse is defined as the ability of a person to obtain a desired outcome from a fixed model (Ustun et al., 2019). In our paper, we aim to fit models that provide *constructive recourse*, i.e. actions that allow decision subjects to improve both their predicted *and* true outcomes.

Our approach may be useful for mitigating the disparate effects of strategic adaptation (Hu et al., 2019; Milli et al., 2019; Liu et al., 2020) that stem from differences in the cost of manipulation (see Proposition 4). Our results may be helpful for developing robust classifiers in dynamic environments, where both decision subjects' features and the deployed models may vary across time periods (Kilbertus et al., 2020; Shavit et al., 2020; Liu & Chen, 2017). Also relevant is the recent work on performative prediction (Perdomo et al., 2020; Miller et al., 2021; Izzo et al., 2021; Mendler-Dünner et al., 2020), in which the choice of model itself affects the distribution over instances. However, this literature differs from ours in that we focus on inducing constructive adaptations from decision subjects at a single step, rather than finding an optimal policy that incurs the minimum deployment error.

Due to page limit, we provide additional related work on strategic classification, algorithmic recourse, causal modeling, and incentive design in Appendix E

## 2 Problem statement

In this section, we describe our approach to training a classifier that incentivizes improving actions.

### 2.1 Preliminaries

We consider a standard classification task of training a classifier $h : \mathbb{R}^d \to \{-1, +1\}$ from a dataset of $n$ examples $\{(x_i, y_i)\}_{i=1}^n$, where example $i$ consists of a vector of $d$ features $x_i \in \mathbb{R}^d$ and a binary label $y_i \in \{-1, +1\}$. Example $i$ corresponds to an agent who wishes to receive a positive prediction $h(x_i) = +1$, and who will alter their features to obtain such a prediction once the model is deployed. We assume that an agent's true qualification (or label), denoted as $y$, is always a function of its feature vector $x$, and define the true unknown qualification function $y : X \to \{0, 1\}$ as the mapping between the feature vector $x \in \mathbb{R}^d$ and the true qualification/label $y \in \{0, 1\}$.

We formalize these dynamics as a sequential game between the following two players: the model designer, and the decision subjects [1]. The objectives for the two players are as follows:

1. A model designer, who trains a classifier $h : \mathcal{X} \to \{-1, +1\}$ from a hypothesis class $\mathcal{H}$, which is also their action space.

2. Decision subjects, who adapt their features from $x$ to $x'$ so as to be assigned $h(x') = +1$ if possible. We assume that decision subjects incur a cost for altering their features, which we represent using a *cost function* $c : \mathcal{X} \times \mathcal{X} \to \mathbb{R}^+$. The action space for the decision subjects includes all feature vectors that are within a given manipulation budget $B$, namely $\forall x' \in \mathbb{R}^d$ such that $c(x, x') \leq B$.

We intentionally do not provide the formal definitions of the utilities for the two players here due to the need to provide a clear and accessible introduction to our framework. We will provide more detailed discussions on the agent and decision maker's utility function in Section 3.2 and Section 2.3, respectively.

We assume that decision subjects know the model designer's classifier, and the model designer knows the decision subjects' cost function. Decision subjects alter their features based on their current features $x$, the cost function $c$, and the classifier $h$, so that their altered features can be written $x_* = \Delta(x; h, c)$ where $\Delta(\cdot)$ is the *best response function*. The model designer only observes the altered feature $x_*$ but not the original and private one $x$ the decision subject holds. In other words, we consider the standard setting in strategic classification where the model designer has no strong verification power to verify truthfulness of $x_*$.

We allow adaptations that alter the true qualification $y$. In practice, the relationship between features and true qualification is unknown, and in fact it is known that distinguishing causal features (features that affect the true outcome) from non-causal features reduces to solving a non-trivial causal inference problem Miller et al. (2020). Addressing this aspect is not the aim of the present work; instead, we will assume that changes in certain features are known to affect the qualification – for example, in loan application, such features can

---

[1]Throughout the paper, we will also use strategic agents, or agents, interchangeably.

be the agent's education level and salary, and changing those features will affect qualification is the agent's ability to pay for the loan.

When an agent adapts its feature vector from $x$ to $\Delta(x)$, its qualification becomes $y(\Delta(x))$, which may differ from $y(x)$. We consider a setting in which during the training process, the decision maker cannot observe how decision subjects' true qualifications change after they alter their features. We introduced the shorthand notation $y$ to refer to $y(x)$, the qualification for the *original* feature vector, for the sake of simplicity. For the rest of our paper, a label $y$ always denotes the true qualification *before* adaptation.

## 2.2 Background

In a standard prediction setting, a model designer trains a classifier that minimizes the *empirical risk*:

$$h^*_{\mathsf{ERM}} \in \arg\min_{h \in \mathcal{H}} R_{\mathsf{ERM}}(h)$$

where $R_{\mathsf{ERM}}(h) = \mathbb{E}_{x \sim \mathcal{D}}[\mathbb{1}(h(x) \neq y)]$. This classifier performs poorly in a setting with strategic adaptation, since the model is deployed on a population with a different distribution over $\mathcal{X}$ (as decision subjects alter their features) and $y$ (as changes in features may alter true outcomes).

Existing approaches in strategic classification tackle these issues by training a classifier that is robust to *all* adaptation. This approach treats all adaptation as undesirable, and seeks to maximize accuracy by discouraging it entirely. Formally, they train a classifier that minimizes the *strategic risk*:

$$h^*_{\mathsf{SC}} \in \arg\min_{h \in \mathcal{H}} R_{\mathsf{SC}}(h)$$

where $R_{\mathsf{SC}}(h) = \mathbb{E}_{x \sim \mathcal{D}}[\mathbb{1}(h(x_*) \neq y)]$, and $x_* = \Delta(x, h; c)$ denotes the features of a decision subject after adaptation. However, this classifier still has suboptimal accuracy because $y$ changes as a result of the adaptation in $x$. Further, this design choice misses the opportunity to encourage a profile $x$ to truly improve to change their $y$.

## 2.3 CA risk: minimizing error while encouraging constructive adaptation

In many applications, model designers are better off when decision subjects adapt their features in a way that yields a specific true outcome, such as $y = +1$. Consider a typical lending application where a model is used to predict whether a customer will repay a loan. In this case, a model designer benefits from $y = +1$, as this means that a borrower will repay their loan.

### 2.3.1 Ideal objective function for the decision maker

Ideally, the decision maker should aim to classify the agents correctly using their adapted features with respect to the corresponding new qualification. Mathematically, this corresponds to training a classifier $h^*$ that minimizes the following quantity:

$$h^* \in \arg\min_{h \in \mathcal{H}} \mathbb{E}_{x \sim \mathcal{D}}[\mathbb{1}(h(\Delta(x)) \neq y(\Delta(x)))] \tag{1}$$

where $\Delta(x)$ is the agent's adapted feature, and $y(\Delta(x))$ is the true qualification after the adaptation. However, since the true mapping function $y : \mathbb{R}^d \to \{0, 1\}$ is unknown, and the decision maker cannot observe how decision subjects' true qualifications change after they alter their features, we need to propose an alternative approach to achieve similar goals of this ideal objective function, which we call $\mathsf{CA}$ risk minimization.

### 2.3.2 Our proposed CA risk

To help explain our proposed approach, we assume that we can write $x = [x_{\mathsf{I}} \mid x_{\mathsf{M}} \mid x_{\mathsf{IM}}]$ where $x_{\mathsf{I}}$, $x_{\mathsf{M}}$ and $x_{\mathsf{IM}}$ denote the following categories of features:

- *Immutable* features ($x_{\mathsf{IM}}$), which cannot be altered (e.g. race, age).

- *Improvable* features ($x_{\mathsf{I}}$), which can be altered in a way that will either increase or decrease the true outcome $y(x)$ (e.g. increasing education level might help improve the probability of repayment).

- *Manipulable* features ($x_{\mathsf{M}}$), which can be altered *without* changing the true outcome $y(x)$ (e.g. social media presence, which can be used as a proxy for influence). Notice that it is the *change* in these features that is undesirable; the features themselves may still be useful for prediction.

**Incomplete taxonomy of features.** There may also be features that can be altered but whose effect is *unknown*. In this work, we treat them as manipulable features. We would like to point out that in practice, implementing our proposed solution does not require the decision-maker to know exactly how to characterize every single feature. In fact, our method can be applied to settings where the decision-makers only know some features are improvable and focus on incentivizing adaptations on them, while treating changes on the rest of the features as undesirable. In this case, using our training method is still strictly better than performing no intervention (i.e. simply letting decision subjects perform their unconstrained best response).

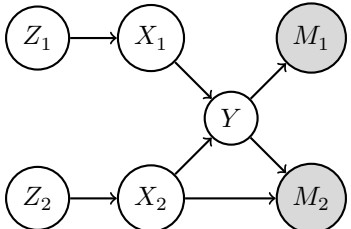

Figure 1: A causal DAG for the `toy` data. $Z_1$ and $Z_2$ are improvable features that determine the true qualification $Y$, $X_1 = Z_1$, and $X_2$ is a noisy proxy for $Z_2$. In our context, all we require is the knowledge that $X_1, X_2$ are the factors that causally affect $Y$, rather than complete knowledge of the DAG. We can directly observe $X_1$ and $X_2$ but not $Z_1$ or $Z_2$. In addition, $M_1$ and $M_2$ are manipulated features that correlate with $Y$.

Please see Figure 1 for a demonstration of the differences between improvable and manipulable features. We also use $x_{\mathsf{A}} = [x_{\mathsf{I}} \mid x_{\mathsf{M}}]$ to denote the *actionable* features, and $d_{\mathsf{A}}$ to denote its dimension. Note that the question of how to decide which features are of which type is beyond the scope of the present work; however, this is the topic of intense study in the causal inference literature (Miller et al., 2020). Analogously, we define the following variants of the best response function $\Delta$:

- $x_*^{\mathsf{I}} = \Delta_{\mathsf{I}}(x, h; c)$: the *improving best response*, which involves an adaptation that only alters improvable features.

- $x_*^{\mathsf{M}} = \Delta_{\mathsf{M}}(x, h; c)$: the *manipulating best response*, which involves an adaptation that only alters manipulable features.

Note that in reality, a decision subject can still alter both types of features, which means that they will perform $\Delta(x, h; c)$, unless the model designer explicitly forbids changing certain features. However, it still worth distinguishing different types of best responses when the model designer designs the classifier: we can think of the improving best response $\Delta_{\mathsf{I}}$ as the best possible adaptation which only consists of honest improvement, while the manipulating best response $\Delta_{\mathsf{M}}$ is the worst possible adaptation that consists of pure manipulation. The model designer would like to design a classifier such that for the decision subjects, $\Delta(x, h; c)$ appears to be close to $\Delta_{\mathsf{I}}(x, h; c)$. We therefore propose to train a classifier that minimizes the *constructive adaptation* (CA) risk $R_{\mathsf{CA}}$, which balances robustness to manipulation and incentivization of improvement:

$$h_{\mathsf{CA}}^* \in \arg\min_{h \in \mathcal{H}} R_{\mathsf{CA}}(h) := R_{\mathsf{M}}(h) + \lambda \cdot R_{\mathsf{I}}(h) \tag{2}$$

The first term, $R_{\mathsf{M}}(h) = \mathbb{E}_{x \sim \mathcal{D}}[\mathbb{1}(h(x_*^{\mathsf{M}}) \neq y)]$, is the *manipulation risk*, which penalizes pure manipulation. The second term, $R_{\mathsf{I}}(h) = \mathbb{E}_{x \sim \mathcal{D}}[\mathbb{1}(h(x_*^{\mathsf{I}}) \neq +1)]$, is the *improvement risk*, which rewards decision subjects for

playing their improving best response. The parameter $\lambda > 0$ trades off between these competing objectives. Setting $\lambda \to 0$ results in an objective that simply discourages manipulation, whereas increasing $\lambda \to \infty$ yields a trivial classifier that always predicts $+1$.

A natural question to ask is: how good the proposed objective function Equation (2) is compared to the ideal objective function in Equation (1)? We show that the two terms in the objective function can be viewed as proxies for the ideal objective function. In particular, in Section 4, we show that under reasonable conditions, the following hold:

- The first term, $R_{\mathsf{M}}(h)$, is an upper bound on $R_{\mathsf{SC}}(h)$. Thus minimizing the manipulation risk also minimizes the traditional strategic risk (Proposition 5).

- A decrease in the second term, $R_{\mathsf{I}}(h)$ reflects an increase in $\Pr(y(x_*^{\mathsf{I}}) = +1)$. Thus improvement in the prediction outcome aligns with improvement in the true qualification (Proposition 6).

## 3 Decision subjects' best response

We now characterize the decision subjects' best response.

### 3.1 Setup

We restrict our analysis to the setting in which a model designer trains a *linear classifier* $h(x) = \text{sign}(w^\top x)$, where $w = [w_0, w_1, \ldots, w_d] \in \mathbb{R}^{d+1}$ denotes a vector of $d + 1$ weights. We capture the cost of altering $x$ to $x'$ through the *Mahalanobis* norm of the changes:[2]

$$c(x, x') = \sqrt{(x_{\mathsf{A}} - x_{\mathsf{A}}')^\top S^{-1} (x_{\mathsf{A}} - x_{\mathsf{A}}')}$$

Here, $S^{-1} \in \mathbb{R}^{d_{\mathsf{A}}} \times \mathbb{R}^{d_{\mathsf{A}}}$ is a symmetric *cost covariance matrix* in which $S_{j,k}^{-1}$ represents the cost of altering features $j$ and $k$ simultaneously. To ensure that $c(\cdot)$ is a valid norm, we require $S^{-1}$ to be *positive definite*, meaning $x_{\mathsf{A}}^\top S^{-1} x_{\mathsf{A}} > 0$ for all $x_{\mathsf{A}} \neq \mathbf{0} \in \mathbb{R}^{d_{\mathsf{A}}}$. Additionally, we assume $S^{-1}$ is a block matrix of the form

$$S^{-1} = \begin{bmatrix} (S^{-1})_{\mathsf{I}} & (S^{-1})_{\mathsf{IM}} \\ (S^{-1})_{\mathsf{MI}} & (S^{-1})_{\mathsf{M}} \end{bmatrix}, \text{or} \ \ S = \begin{bmatrix} S_{\mathsf{I}} & S_{\mathsf{IM}} \\ S_{\mathsf{MI}} & S_{\mathsf{M}} \end{bmatrix} \tag{3}$$

Notice that the $I$-th block of matrix $S^{-1}$ (i.e. $(S^{-1})_{\mathsf{I}}$) does not necessarily equal to its inverse's $I$-th block component (i.e. $S_{\mathsf{I}}^{-1}$).

We allow the cost matrix to contain non-zero elements on non-diagonal entries. This means that our results hold even when there are interaction effects when altering multiple features. This generalizes prior work on strategic classification in which the cost is based on the $\ell_2$ norm of the changes, which is tantamount to setting $S^{-1} = I$, and therefore assumes the change in each feature contributes independently to the overall cost (see e.g., Hardt et al., 2016a; Haghtalab et al., 2020).

### 3.2 Decision subject's best response model

Given the assumptions of Section 3.1, we can define and analyze the decision subjects' best response. We start by defining the decision subject's payoff function. Given a classifier $h$, a decision subject who alters their features from $x$ to $x'$ derives total utility
$$U(x, x') = h(x') - c(x, x')$$

Naturally, a decision subject tries to maximize their utility; that is, they play their *best response*:

**Definition 3.1** (F-Best Response Function). *Let $\mathsf{F} \in \{\mathsf{I}, \mathsf{M}, \mathsf{A}\}$, and let $\mathcal{X}_{\mathsf{F}}^*(x)$ denote the set of vectors that differ from $x$ only in features of type $\mathsf{F}$. Let $\Delta_{\mathsf{F}} : \mathcal{X} \to \mathcal{X}$ denote the $\mathsf{F}$-best response of a decision subject with features $x$ to $h$, defined as:*
$$\Delta_{\mathsf{F}}(x) = \arg\max_{x' \in \mathcal{X}_{\mathsf{F}}^*(x)} U(x, x')$$

---

[2]Since immutable features $x_{\mathsf{IM}}$ cannot be altered, the cost function involves only the actionable features $x_{\mathsf{A}}$.

Setting $\mathsf{F} = \mathsf{I}$ gives the *improving best response* $\Delta_\mathsf{I}(x)$, in which the adaptation changes only the improvable features; setting $\mathsf{F} = \mathsf{M}$ yields the *manipulating best response* $\Delta_\mathsf{M}(x)$, in which only manipulable features are changed. Setting $\mathsf{F} = \mathsf{A}$, we get the standard *unconstrained best response* $\Delta_\mathsf{A}(x)$ in which any actionable features can be changed. As we mentioned earlier, we will also use $x_*^\mathsf{F} := \Delta_\mathsf{F}(x)$ as shorthand for the $\mathsf{F}$-best response, and we denote $\Delta(x) := \Delta_\mathsf{A}(x)$.

Intuitively, the cost of manipulation should be smaller than the cost of actual improvement. For example, improving one's coding skills should take more effort, and thus be more costly, than simply memorizing answers to coding problems. As a result, one would expect the gaming best response $\Delta_\mathsf{M}(x)$ and the unconstrained best response $\Delta(x)$ to flip a negative decision more easily than the improving best response $\Delta_\mathsf{I}(x)$. In Section 3.3, we formalize this notion (Proposition 2).

For ease of notation, let $\widehat{S}_\mathsf{F} := ((S^{-1})_\mathsf{F})^{-1}$. We prove the following theorem characterizing the decision subject's different best responses:

**Theorem 1** ($\mathsf{F}$-Best Response in Closed-Form). *Given a linear threshold function $h(x) = \text{sign}(w^\mathsf{T} x)$ and a decision subject with features $x$ such that $h(x) = -1$, reorder the features so that $x = [x_\mathsf{F} \mid x_{\mathsf{A} \backslash \mathsf{F}} \mid x_\mathsf{IM}]$, and let $\Omega_\mathsf{F} = w_\mathsf{F}^\mathsf{T} \widehat{S}_\mathsf{F} w_\mathsf{F}$. Then $x$ has $\mathsf{F}$-best response*

$$\Delta_\mathsf{F}(x) = \begin{cases} \left[ x_\mathsf{F} - \frac{w^\mathsf{T} x}{\Omega_\mathsf{F}} \widehat{S}_\mathsf{F} w_\mathsf{F} \right] \mid x_{\mathsf{A} \backslash \mathsf{F}} \mid x_\mathsf{IM}, & \text{if } \frac{|w^\mathsf{T} x|}{\sqrt{\Omega_\mathsf{F}}} \leq 2 \\ x, & \text{otherwise} \end{cases} \tag{4}$$

*with corresponding cost*

$$c(x, \Delta_\mathsf{F}(x)) = \begin{cases} \frac{|w^\mathsf{T} x|}{\sqrt{\Omega_\mathsf{F}}}, & \text{if } \frac{|w^\mathsf{T} x|}{\sqrt{\Omega_\mathsf{F}}} \leq 2 \\ 0 & \text{otherwise} \end{cases}.$$

All proofs in this section are included in Appendix B.

*Example:* When $\mathsf{F} = \mathsf{M}$, $x_\mathsf{F} = x_\mathsf{M}$ and $x_{\mathsf{A} \backslash \mathsf{F}} = [x_\mathsf{I}]$. After reordering features, we get the following closed-form expression for the manipulating best response:

$$\Delta_\mathsf{M}(x) = \begin{cases} \left[ x_\mathsf{I} \mid x_\mathsf{M} - \frac{w^\mathsf{T} x}{\Omega_\mathsf{M}} \widehat{S}_\mathsf{M} w_\mathsf{M} \mid x_\mathsf{IM} \right] & \text{if } \frac{|w^\mathsf{T} x|}{\sqrt{\Omega_\mathsf{M}}} \leq 2 \\ x, & \text{otherwise} \end{cases}$$

with corresponding cost

$$c(x, \Delta_\mathsf{M}(x)) = \begin{cases} \frac{|w^\mathsf{T} x|}{\sqrt{\Omega_\mathsf{M}}}, & \text{if } \frac{|w^\mathsf{T} x|}{\sqrt{\Omega_\mathsf{M}}} \leq 2 \\ 0 & \text{otherwise} \end{cases}.$$

### 3.3 Discussion

We now discuss the implications of different decision subject's responses derived in Theorem 1. In this section, we consider a slightly more structured cost matrix that is diagonal blocked matrix (in which case, $S_\mathsf{IM}^{-1} = S_\mathsf{MI}^{-1} = \mathbf{0}$), which corresponds to a setting where there are no correlations between the *cost* of changing manipulated feature versus the cost of changing improvable features. We include the proofs in Appendix C.

Firstly, we demonstrate a basic limitation for the model designer: if the classifier uses any manipulable features as predictors, then decision subjects will find a way to exploit them. Hence the only way to avoid any possibility of manipulation is to train a classifier without such features.

**Proposition 1** (Preventing Manipulation is Hard). *Suppose there exists a manipulated feature $x^{(m)}$ whose weight in the classifier $w_\mathsf{A}^{(m)}$ is nonzero. Then for almost every $x \in \mathcal{X}$, $\Delta^{(m)}(x) \neq x^{(m)}$.* [3]

---

[3] In our paper, the subscript (e.g. $x_m$) refers to the entire feature vector (e.g., $x_m \in R^{d_m}$, where $d_m$ is the total number of the manipulative features), while the superscript $(m)$ refers to the particular index of a particular manipulation feature.

Next, we show that the unconstrained best response $\Delta(x)$ dominates the improving best response $\Delta_\mathsf{I}(x)$, thus highlighting the difficulty of inducing decision subjects to change only their improvable features when they are also allowed to change manipulable features.

**Proposition 2** (Unconstrained Best Response Dominates Improving Best Response). *Suppose there exists a manipulable feature $x^{(m)}$ whose weight in the classifier $w_A^{(m)}$ is nonzero. Then, if a decision subject can flip her decision by playing the improving best response, she can also do so by playing the unconstrained best response. The converse is not true: there exist decision subjects who can flip their predictions through their unconstrained best response but not their improving best response.*

Next, we show how correlations between features affect the cost of adaptation. This can be demonstrated by looking at any cost matrix and adding a small nonzero quantity $\tau$ to some $i, j$-th and $j, i$-th entries. Such a perturbation can reduce every decision subject's best-response cost:

**Proposition 3** (Correlations between Features May Reduce Cost). *For any cost matrix $S^{-1}$ and any nontrivial classifier $h$, there exist indices $k, \ell \in [d_A]$ and $\tau \in \mathbb{R}$ such that every feature vector $x$ has lower best-response cost under the cost matrix $\tilde{S}^{-1}$ given by*

$$\tilde{S}_{ij}^{-1} = \tilde{S}_{ji}^{-1} = \begin{cases} S_{ij}^{-1} + \tau, & \text{if } i = k, j = \ell \\ S_{ij}^{-1}, & \text{otherwise} \end{cases}$$

*than under $S^{-1}$; that is, $c_{\tilde{S}^{-1}}(x, \Delta(x)) < c_{S^{-1}}(x, \Delta(x))$ for all $x$.*

In many applications, decision subjects may incur different costs for modifying their features, resulting in disparities in prediction outcomes (see Hu et al., 2019, for a discussion). To formalize this phenomenon, suppose $\Phi$ and $\Psi$ are two groups whose costs of changing improvable features are identical, but members of $\Phi$ incur higher costs for changing manipulable features. Let $\phi \in \Phi$ and $\psi \in \Psi$ be two people from these groups who share the same profile, i.e. $x_\phi = x_\psi$. We show the following:

**Proposition 4** (Cost Disparities between Subgroups). *Suppose there exists a manipulated feature $x^{(m)}$ whose corresponding weight in the classifier $w_A^{(m)}$ is nonzero. Then if decision subjects are allowed to modify any features, $\phi$ must pay a higher cost than $\psi$ to flip their classification decision.*

Proposition 4 highlights the importance for a model designer to account for these differences when serving a population with heterogeneous subgroups. Indeed, when one group achieves more favorable prediction outcomes due to a lower cost of manipulation, our method mitigates the cost disparities between different subgroups by encouraging changes in improvable features and penalizing manipulation.

## 4 Constructive adaptation risk minimization

In this section we analyze the training objective for the model designer, formulating it as an empirical risk minimization (ERM) problem. Any omitted details can be found in Appendix D.

The model designer's goal is to publish a classifier $h$ that maximizes the classification accuracy while incentivizing individuals to change their improvable features. By Theorem 1, we have

$$x_*^\mathsf{M} = \begin{cases} \left[x_\mathsf{I} \mid x_\mathsf{M} - \frac{w^\top x}{\Omega_\mathsf{M}} \widetilde{S}_\mathsf{M} w_\mathsf{M} \mid x_\mathsf{IM}\right] & \text{if} \frac{|w^\top x|}{\sqrt{\Omega_\mathsf{M}}} \leq 2 \\ x, & \text{otherwise} \end{cases} \tag{5}$$

$$x_*^\mathsf{I} = \begin{cases} \left[x_\mathsf{I} - \frac{w^\top x}{\Omega_\mathsf{I}} \widetilde{S}_\mathsf{I} w_\mathsf{I} \mid x_\mathsf{M} \mid x_\mathsf{IM}\right], & \text{if} \frac{|w^\top x|}{\sqrt{\Omega_\mathsf{I}}} \leq 2 \\ x, & \text{otherwise} \end{cases} \tag{6}$$

Recall from Section 2.3 that the model designer's optimization program is as follows:

$$\min_{h \in \mathcal{H}} \quad \mathbb{E}_{x \sim \mathcal{D}} \left[\mathbb{1}\left(h(x_*^\mathsf{M}) \neq y\right)\right] + \lambda \mathbb{E}_{x \sim \mathcal{D}} \left[\mathbb{1}\left(h(x_*^\mathsf{I}) \neq +1\right)\right]$$
$$\text{s.t.} \quad x_*^\mathsf{M} \text{ in Eq. (5)}, \ x_*^\mathsf{I} \text{ in Eq. (6)} \tag{7}$$

**Interpreting the objective.** The two terms in the objective function can be viewed as proxies for two other familiar objectives. The first term, $\mathbb{E}_{x \sim \mathcal{D}}\left[\mathbb{1}(h(x_*^{\mathsf{M}}) \neq y)\right]$, directly penalizes pure manipulation. But as the following proposition suggests, minimizing this term also minimizes the traditional strategic risk when the true qualification does not change:

**Proposition 5.** *Assume that the manipulating best response is more likely to result in a positive prediction than the unconstrained best response, given that the true labels do not change. Then*

$$\mathbb{E}_{x \sim \mathcal{D}}\left[\mathbb{1}[h(x_*) \neq y] \mid \Delta(y) = y\right] \leq \mathbb{E}_{x \sim \mathcal{D}}\left[\mathbb{1}(h(x_*^{\mathsf{M}}) \neq y)\right].$$

Intuitively, the assumption within Proposition 5 may be fulfilled in settings where a population of agents each have the same fixed budget on the cost or effort they are willing to expend, and manipulative or cheating-type actions (for instance, (controlling recent purchase behaviors and borrowing money from family members right before applying for a credit card) confer greater immediate advantages than honest improvement (e.g. spending frugally and accruing savings from personal income over several years).

The second term, $\mathbb{E}_{x \sim \mathcal{D}}\left[\mathbb{1}(h(x_*^{\mathsf{I}}) \neq +1)\right]$, explicitly rewards decision subjects for playing their improving best response (closely related to the notion of *recourse*). Of course, without positing a causal graph, we cannot know whether performing the improving best response leads to a positive change in the true qualification, namely whether $\Delta_{\mathsf{I}}(Y) = +1$; however, when the distribution of $X$ may change but not the conditional label distribution $\Pr(Y|X)$, we can show that an increase in $\Pr(h(X) = +1)$ reflects an increase in $\Pr(Y = +1)$. This gives formal evidence that our prediction outcome aligns with improvement in the true qualification:

**Proposition 6.** *Let $\mathcal{D}^*$ be the new distribution after decision subject's best response. Denote $\omega_h(x) = \frac{\Pr_{\mathcal{D}^*}(X=x)}{\Pr_{\mathcal{D}}(X=x)}$ denote the amount of adaptation induced at feature vector $x$. Suppose $y(X)$ and $h(X)$ are both positively correlated with $\omega_h(X)$, and that the distribution of the true label $Y$ given a particular feature vector $X$ is unchanged is the same before and after adaptation. Then the following are equivalent:*

$$\Pr[h(x_*^{\mathsf{I}}) = +1] > \Pr[h(x) = +1] \iff \Pr[y(x_*^{\mathsf{I}}) = +1] > \Pr[y(x) = +1].$$

Proofs of Propositions 5 and 6 can be found in Appendix D.1 and D.2. We also provide further derivation for model designer's objective function in Appendix D.3.

Here we provide some motivation for the premise of Proposition 6. An unchanged $\Pr(Y|X)$ means that the mapping from feature vector $X$ to its corresponding true qualification $Y(X)$ remains the same despite a population-level distribution shift. This is a useful and natural simplification in numerous settings. An example is in credit card applications: suppose $X$ is an applicant's credit score and $Y$ is whether they are truly qualified. For people with the same credit score, we assume they have equal chances of being truly qualified.

---

**Algorithm 1** Best Response for Non-Linear Model

---

**Input:** Non-Linear classifier $h$, an individual data point $x$
**Result:** $x_*^{\mathsf{M}}$ and $x_*^{\mathsf{I}}$
**Step 1.** Call LIME to get the approximated weights $\tilde{w}$ of a local linear classifier for non-linear model $h$ around the individual point $x$
**Step 2.** Substitute $\tilde{w}$ into Eq. (5) and Eq. (6) to get $x_*^{\mathsf{M}}$ and $x_*^{\mathsf{I}}$, respectively

---

**Extension to non-linear models.** The above approach in Eq. (7) presumes a linear classifier such that we can derive a close-form solution of the agent's best response. However, the recourse scheme will be typically infeasible with non-linear classifiers. To extend our approach to nonlinear models, we propose to substitute $x_*^{\mathsf{M}}$ and $x_*^{\mathsf{I}}$ in Eq. (7) with an approximated best response acquired from a local linear classifier. We note that a prior work LIME (Ribeiro et al., 2016) can provide an approximate linear decision boundary for arbitrary individual points to any non-linear models. The idea is to sample the spherical neighborhood of the data point and fit a local linear model with the target model's certified predictions. As shown in Algorithm 1,

we integrate LIME into the oracle that can return us any decision subjects' best response in terms of the approximated local linear classifier. Once we get the best response $x_*^{\mathsf{M}}$ and $x_*^{\mathsf{I}}$, we iteratively plug them back to Eq. (7) as the learning objective of the non-linear classifier. We will demonstrate the effectiveness of this oracle procedure when optimizing a non-linear neural network with gradient descent in Appendix F.5. Nonetheless, even with the above extension, all of our theoretical guarantees is not straightforwardly clear to analysis with an oracle of non-linear models' best response, so we let the current paper focus on linear models.

## 5 Experiments

In this section, we present empirical results to benchmark our proposed method on synthetic and real-world datasets. We test the effectiveness of our approach in terms of its ability to incentivize improvement as well as to disincentivize manipulation (see **Evaluation Criteria** for details). We also compare its performance with other standard approaches (see **Methods**). Our submission includes all datasets, scripts, and source code used to reproduce the results in this section.

### 5.1 Setup

**Datasets and Cost Matrix.** We consider five datasets:

`toy`, a synthetic dataset based on the causal DAG in Fig. 1; `credit`, a dataset for predicting whether an individual will default on an upcoming credit payment (Yeh & Lien, 2009); `adult`, a census-based dataset for predicting adult annual incomes; `german`, a dataset to assess credit risk in loans; and `spambase`, a dataset for email spam detection. The last three are from the UCI ML Repository (Dua & Graff, 2017). We provide a detailed description of each dataset along with a partitioning of features in Table 3 in the Appendix.

We assume the cost of manipulation is lower than that of improvement and refer the specific cost matrix $S$ to Appendix F.2; in particular, we specify the cost matrix $S$ as follows: use cost matrices $S_{\mathsf{I}}^{-1} = I$ and $S_{\mathsf{M}}^{-1} = 0.2I$. We also provide results for non-diagonal cost matrix in the Appendix F.4.

$$S_{ij}^{-1} = \begin{cases} 1, & \text{if } i = j \text{ and } i \in \mathsf{I} \\ 0.2, & \text{if } i = j \text{ and } j \in \mathsf{M} \\ 1, & \text{if the cost of changing features } i \\ & \text{and } j \text{ are } \textit{negatively} \text{ correlated} \\ -1, & \text{if the cost of changing features } i \\ & \text{and } j \text{ are } \textit{positively} \text{ correlated} \\ 0, & \text{otherwise} \end{cases}$$

We use the `credit` dataset as a demonstration of how we specify the non-diagonal element in the cost matrix. For two feature variables that have a positive correlation, e.g., *CheckingAccountBalance* and *SavingsAccountBalance*, we assign $-1$ to the corresponding elements in the cost matrix $S$. For two feature variables that have a negative correlation, e.g., *CheckingAccountBalance* and *MissedPayments*, we assign $+1$ to the corresponding elements in the cost matrix. In practice, the cost matrix $S$ should be determined using domain expertise. The purpose of the cost matrix used in these experiments is not to accurately specify costs per se, but to demonstrate the relative difficulty of changing different features.

**Methods.** We fit linear classifiers for each dataset using the following methods: `ST`, a static classifier trained using $\ell_2$-logistic regression without accounting for strategic adaptation; `DF`, a classifier trained using $\ell_2$-logistic regression without any manipulated features; `MP`, a classifier that considers the agent's unconstrained best response (i.e. with changes in any actionable features $x_{\mathsf{A}}$ allowed) during training, as typically done in the strategic classification literature (Hardt et al., 2016a); `CA`, a linear logistic regression classifier that results from solving the optimization program in Eq. (23), which is a smooth differentiable surrogate version of the objective function Eq. (7). Please refer to Appendix D.3 for a detailed derivation. Using the BFGS algorithm (Byrd et al., 1995). `CA` represents our approach.

**Evaluation Criteria.** We run each method with 5-fold cross-validation and report the following:

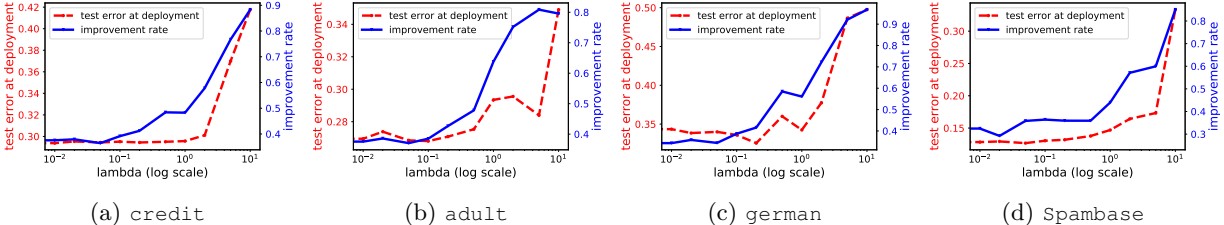

(a) credit    (b) adult    (c) german    (d) Spambase

Figure 2: We plot the trade-off between test error at deployment and improvement rate in terms of cost matrix. We observe that the test error increases consistently with the increase of the improvement rate.

Table 1: Performance metrics for different specifications (**Spec.**) in which features may be misspecified. For each method, we report *test error*, *deployment error*, and *improvement rate*. In Full, the model designer has full knowledge of the causal DAG. In Mis. I, $M_1$ is mistaken for an improvable feature. In Mis. II, the improvable feature $X_1$ is miscategorized as manipulable.

| | | METHODS | | | |
|---|---|---|---|---|---|
| **Spec.** | **Metrics** | ST | DF | MP | CA |
| Full | *test error* | 10.29 | 28.0 | 11.91 | 11.62 |
| | *deployment error* | 35.79 | 35.15 | 24.1 | 20.61 |
| | *improvement rate* | 11.54 | 13.13 | 14.63 | 23.49 |
| Mis. I | *test error* | 11.39 | 10.52 | 11.26 | 11.04 |
| | *deployment error* | 37.37 | 10.53 | 19.79 | 25.30 |
| | *improvement rate* | 37.23 | 39.74 | 0.62 | 23.04 |
| Mis. II | *test error* | 10.58 | 35.77 | 29.52 | 10.80 |
| | *deployment error* | 12.37 | 41.51 | 27.68 | 23.58 |
| | *improvement rate* | 1.12 | 5.74 | 3.36 | 19.82 |

- *Test Error*: the error of a classifier after training but *before* decision subjects' adaptations, i.e. $\mathbb{E}_{(x,y)\sim\mathcal{D}}\,\mathbb{1}[h(x) \neq y]$.

- *(Worst-Case) Deployment Error*: the test error of a classifier *after* decision subjects play their manipulating best response, i.e. $\mathbb{E}_{(x,y)\sim\mathcal{D}}\,\mathbb{1}[h(x^{\mathsf{M}}_*) \neq y]$.

- *(Best-Case) Improvement Rate*: the percent of improvement, defined as the proportion of the population who originally would be rejected but are accepted if they perform constructive adaptation (improving best response), i.e. $\mathbb{E}_{(x,y)\sim\mathcal{D}}\,\mathbb{1}[h(x^{\mathsf{I}}_*) = +1 \mid y(x) = -1]$.

## 5.2  Controlled experiments on synthetic dataset

We perform controlled experiments using a synthetic `toy` dataset to test the effectiveness of our model at incentivizing improvement in various situations. As shown in Fig. 1, we set $Z_1$ and $Z_2$ as improvable features, $X_1$ and $X_2$ as their corresponding noisy proxies, $M_1$ and $M_2$ as manipulable features, and $Y$ as the true outcome. Since we have full knowledge of this DAG structure, we can observe the changes in the true outcome after the decision subject's best response. As shown in Table 1, Our method achieves the lowest deployment error (20.61%) and the best improvement rate (23.04%) when the model designer has full knowledge of the causal graph.

We also run experiments in which some features are *misspecified*, simulating realistic scenarios in which the model designer may not be able to observe all the improvable features (Haghtalab et al., 2020; Shavit et al., 2020), or mistakes one type of feature for another. We model these situations by changing $M_1$ into an improvable feature and $X_1$ into a manipulable feature; the results, shown in Table 1, show that our classifier maintains a relatively high improvement rate in these cases, without sacrificing much deployment accuracy.

Table 2: Performance metrics for all methods over 4 real data sets with non-diagonal cost matrix. We report the mean and standard deviation for 5-fold cross validation. The constructive adaptation (CA) consistently achieves a high accuracy at deployment while providing the highest improvement rates across all four datasets.

| Dataset | Metrics | METHODS | | | |
|---------|---------|---------|---------|---------|---------|
| | | ST | DF | MP | CA |
| CREDIT | *test error* | $29.52 \pm 0.37$ | $29.66 \pm 0.40$ | $29.65 \pm 0.41$ | $29.60 \pm 0.44$ |
| | *deploy error* | $31.25 \pm 0.56$ | $29.66 \pm 0.40$ | $29.41 \pm 0.32$ | $29.49 \pm 0.38$ |
| | *improvement rate* | $46.35 \pm 3.81$ | $44.71 \pm 4.75$ | $36.76 \pm 0.53$ | $48.27 \pm 5.50$ |
| ADULT | *test error* | $23.05 \pm 0.47$ | $33.55 \pm 0.73$ | $24.94 \pm 0.52$ | $27.22 \pm 0.65$ |
| | *deploy error* | $38.64 \pm 4.46$ | $33.55 \pm 0.73$ | $26.85 \pm 0.59$ | $29.34 \pm 0.45$ |
| | *improvement rate* | $30.92 \pm 3.31$ | $60.63 \pm 29.40$ | $36.70 \pm 1.62$ | $63.79 \pm 7.80$ |
| GERMAN | *test error* | $30.85 \pm 0.82$ | $36.10 \pm 1.97$ | $33.25 \pm 1.44$ | $34.70 \pm 2.15$ |
| | *deploy error* | $33.40 \pm 1.78$ | $36.10 \pm 1.97$ | $34.60 \pm 1.94$ | $34.25 \pm 1.78$ |
| | *improvement rate* | $41.20 \pm 5.77$ | $42.10 \pm 9.07$ | $33.50 \pm 2.53$ | $56.10 \pm 6.40$ |
| SPAMBASE | *test error* | $7.11 \pm 0.52$ | $10.18 \pm 0.45$ | $11.52 \pm 0.12$ | $14.37 \pm 0.24$ |
| | *deploy error* | $22.40 \pm 3.14$ | $10.18 \pm 0.45$ | $12.92 \pm 0.58$ | $14.70 \pm 0.36$ |
| | *improvement rate* | $40.04 \pm 13.06$ | $32.46 \pm 14.63$ | $26.42 \pm 4.80$ | $43.98 \pm 6.18$ |

### 5.3 Results

We summarize the performance of each method in Table 4. To wrap up, our method produces classifiers that achieve almost the highest deployment accuracy while providing the highest percentage of improvement across all four datasets. The static classifier, which does not account for adaptations, is vulnerable to strategic manipulation and consequently has the highest deployment error on every dataset. Naively cutting off the manipulated features may harm the accuracy at test time – DF incurs high test errors on Adult (33.55%) and German (36.10%). In particular, the strategic classifier MP induces the lowest improvement rates on the Credit (36.76%) and German (34.50%) datasets.

**Effect of trade-off parameter $\lambda$.** Fig. 2 shows the performance of linear classifiers for different values of $\lambda$ on four real datasets. Note that, since the objective function is non-convex, the trends for test error at deployment are not necessarily monotonic. In general, we observe a trade-off between the improvement rate and deployment error: both increase as $\lambda$ increases from 0.01 to 10 in all four datasets.

## 6 Conclusion

In this work, we study how to train a linear classifier that encourages constructive adaption. We characterize the equilibrium behavior of both the decision subjects and the model designer, and prove other formal statements about the possibilities and limits of constructive adaptation. Finally, our empirical evaluations demonstrate that classifiers trained via our method achieve favorable trade-offs between predictive accuracy and inducing constructive behavior. Our work has several limitations:

1. As a first foray into strategic classification with constructive adaptation, our focus on linear threshold classifiers helps us capture the challenges unique to this setting; indeed, this is ultimately what allows for a closed-form best response (Theorem 1) even with a significantly more general cost function than in preceding literature. However, this is clearly not true of many models actually in deployment.

2. In order to focus on the *strategic* aspects of constructive adaptation, we assume that the feature taxonomy is simply given; however, distinguishing improvable features from non-improvable features is an interesting question in its own right, and has been shown to be reducible to a nontrivial causal inference problem (Miller et al., 2020).

3. In real-world scenarios, causal features are often intertwined with non-causal features, and improving one may affect the other. While in our paper, we simplify the setting by assuming independence between the effects, we acknowledge that this is not always the case in practice. One potential way to address this issue is to incorporate additional modeling techniques that account for the causal interactions between features, such as causal inference methods or structural equation modeling.

**Boarder Impact.** Since our method incentivizes people to behave in a certain way, to ensure it works fairly and accurately in practice, it should be paired with a rigorous study of the causal relationship between features to decide which are improvable versus manipulable.

### Acknowledgments

This work is partially supported by the National Science Foundation (NSF) under grants IIS-2143895 and IIS-2040800.

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

# Appendix

## A  Organization of the Appendix

The Appendix is organized as follows.

- Section A provides the organization of the appendix.
- Section B provides the proof of Theorem 1.
- Section C includes notations and proofs for the discussion in section 3.3.
- Section D includes the proofs and derivations for section 4.
- Section E presents additional related works.
- Section F shows additional experimental details and results, including basic information on each dataset and the computing infrastructure.

## B  Proof of Theorem 1

In this section, we provide the proof of Theorem 1. To simplify our discussion, we focus on the unconstrained best response, i.e. the case in which $\mathsf{F} = \mathsf{A}$. The proofs for the other two types of best response ($\mathsf{F} = \mathsf{M}$, $\mathsf{F} = \mathsf{I}$) follow the same arguments except that the inverse of $(S^{-1})_{\mathsf{I}}$ does not equal to $S$, but equals to $((S^{-1})_{\mathsf{I}})^{-1}$.

We first prove two lemmas that allow us to reformulate the best response as an optimization problem. The first states that the decision subject's goal is to maximize their utility, but they are unwilling to pay a cost greater than 2:

**Lemma 1** (Decision Subject's Best-Response Function). *Given a classifier $h : \mathcal{X} \to \{-1, +1\}$, a cost function $c : \mathcal{X} \times \mathcal{X} \to \mathbb{R}$, and a set of realizable feature vectors $\mathcal{X}^{\dagger} \subseteq \mathcal{X}$, the* best response *of a decision subject with features $x \in \mathcal{X}^{\dagger}$ is the solution to the following optimization program:*

$$\max_{x' \in \mathcal{X}^{\dagger}} \quad U(x, x') \quad \text{s.t.} \quad c(x, x') \leq 2$$

*Proof.* Since the classifier in our game outputs a binary decision ($-1$ or $+1$), decision subjects only have an incentive to change their features from $x$ to $x'$ when $c(x, x') \leq 2$. To see this, notice that an decision subject originally classified as $-1$ receives a default utility of $U(x, x) = f(x) - 0 = -1$ by presenting her original features $x$. Since costs are always non-negative, she can only hope to increase her utility by flipping the classifier's decision. If she changes her features to some $x'$ such that $f(x') = +1$, then the new utility will be given by

$$U(x, x') = f(x') - c(x, x') = 1 - c(x, x')$$

Hence the decision subject will only change her features if $1 - c(x, x') \geq f(x) = -1$, or $c(x, x') \leq 2$. $\qquad \square$

The next lemma turns the above maximization program into a minimization program, in which the decision subject seeks the minimum-cost change in $x$ that crosses the decision boundary. If the cost exceeds 2, which is the maximum possible gain from adaptation, they would rather not modify any features.

**Lemma 2.** *Let $x^{\star}$ be an optimal solution to the following optimization problem:*

$$x^{\star} = \arg\min_{x' \in \mathcal{X}_{\mathsf{A}}^{*}(x)} \ c(x, x')$$

$$\text{s.t.} \quad \text{sign}(w^{\mathsf{T}} x') = 1$$

*If no solution is returned, we say an $x^{\star}$ such that $c(x, x^{\star}) = \infty$ is returned. Define $\Delta(x)$ as follows:*

$$\Delta(x) = \begin{cases} x^{\star}, & \text{if } c(x, x^{\star}) \leq 2 \\ x, & \text{otherwise} \end{cases}$$

*Then $\Delta(x)$ is an optimal solution to the optimization problem in Lemma 1.*

*Proof.* Recall that the utility function of the decision subject is $U(x, x') = f(x') - c(x, x')$, and that, by Lemma 1, they will only modify their features if the utility increases, i.e. if they achieve $f(x') = +1$ and while incurring cost $c(x, x') \leq 2$.

Consider two cases for $x' \neq x$:

1. When $c(x, x') > 2$, there are no feasible points for the optimization problem of Lemma 1.

2. When $c(x, x') \leq 2$, we only need to consider those feature vectors $x'$ that satisfy $f(x') = 1$, because if $f(x') = -1$, the decision subject with features $x$ would prefer not to change anything. Since maximizing $U(x, x') = f(x') - c(x, x')$ is equivalent to minimizing $c(x, x')$ if $f(x') = 1$, we conclude that when $c(x, x') \leq 2$, the optimum of the program of Lemma 1 is the same as the optimum of the program in Lemma 2.

$\square$

Lemma 2 enables us to re-formulate the objective function as follows. Recall that $c(x, x') = \sqrt{(x_A - x_A')^\mathsf{T} S^{-1} (x_A - x_A')}$ where $S^{-1}$ is symmetric positive definite. Thus $S^{-1}$ has the following diagonalized form, in which $Q$ is an orthogonal matrix and $\Lambda^{-1}$ is a diagonal matrix:

$$S^{-1} = Q^\mathsf{T} \Lambda^{-1} Q = (\Lambda^{-\frac{1}{2}} Q)^\mathsf{T} (\Lambda^{-\frac{1}{2}} Q)$$

With this, we can re-write the cost function as

$$
\begin{aligned}
c(x, x') &= \sqrt{(x_A - x_A')^\mathsf{T} S^{-1} (x_A - x_A')} \\
&= \sqrt{(x_A - x_A')^\mathsf{T} (\Lambda^{-\frac{1}{2}} Q)^\mathsf{T} (\Lambda^{-\frac{1}{2}} Q)(x_A - x_A')} \\
&= \sqrt{(\Lambda^{-\frac{1}{2}} Q(x_A - x_A'))^\mathsf{T} (\Lambda^{-\frac{1}{2}} Q(x_A - x_A'))} \\
&= \|\Lambda^{-\frac{1}{2}} Q(x_A - x_A')\|_2
\end{aligned}
$$

Meanwhile, the constraint in Lemma 2 can be written

$$
\begin{aligned}
\operatorname{sign}(w \cdot x') &= \operatorname{sign}(w_A \cdot x_A' + w_{IM} \cdot x_{IM}) \\
&= \operatorname{sign}(w_A \cdot x_A' - (-w_{IM} \cdot x_{IM})) = 1
\end{aligned}
$$

Hence the optimization problem can be reformulated as

$$\min_{x_A' \in \mathcal{X}_A^*} \|(\Lambda^{-\frac{1}{2}} Q(x_A - x_A'))\|_2 \tag{8}$$

$$\text{s.t. } \operatorname{sign}(w_A \cdot x_A' - (-w_{IM} \cdot x_{IM})) = 1 \tag{9}$$

The above optimization problem can be further simplified by getting rid of the $\operatorname{sign}(\cdot)$:

**Lemma 3.** *If $x_A^\mp$ is an optimal solution to Eq.* (8) *under constraint Eq.* (9)*, then it must satisfy $w_A \cdot x_A^\mp - (-w_{IM} \cdot x_{IM}) = 0$.*

*Proof.* We prove by contradiction. Let $x_A^\mp$ is an optimal solution to Eq. (8) and suppose towards contraction that $w_A x_A^\mp > -w_{IM} \cdot x_{IM}$. Since the original feature vector $x$ was classified as $-1$, we have

$$w_A \cdot x_A^\mp > -w_{IM} \cdot x_{IM}, \quad w_A \cdot x_A < -w_{IM} \cdot x_{IM}$$

By the continuity properties of linear vector space, there exists $\mu \in (0, 1)$ such that:

$$w_\mathsf{A}\left(\mu \cdot x_\mathsf{A}^{\mp} + (1 - \mu)x_\mathsf{A}\right) = -w_\mathsf{IM} \cdot x_\mathsf{IM}$$

Let $x_\mathsf{A}'' = \mu \cdot x_\mathsf{A}^{\mp} + (1 - \mu)x_\mathsf{A}$. Then $\mathrm{sign}(w_\mathsf{A}x_\mathsf{A}'' - (-w_\mathsf{IM} \cdot x_\mathsf{IM})) = 1$, i.e., $x_\mathsf{A}''$ also satisfies the constraint. Since $x_\mathsf{A}^{\mp}$ is an optimum of Eq. (8), we have

$$\|\Sigma^{-\frac{1}{2}}Q(x_\mathsf{A}^{\mp} - x_\mathsf{A})\| \le \|\Sigma^{-\frac{1}{2}}Q(x_\mathsf{A}'' - x_\mathsf{A})\|$$

However, we also have:

$$\begin{aligned}
\|\Sigma^{-\frac{1}{2}}Q(x_\mathsf{A}'' - x_\mathsf{A})\| &= \|\Sigma^{-\frac{1}{2}}Q(\mu \cdot x_\mathsf{A}^{\mp} + (1 - \mu)x_\mathsf{A} - x_\mathsf{A})\| \\
&= \|\Sigma^{-\frac{1}{2}}Q(\mu \cdot (x_\mathsf{A}^{\mp} - x_\mathsf{A}))\| \\
&= \mu\|\Sigma^{-\frac{1}{2}}Q(x_\mathsf{A}^{\mp} - x_\mathsf{A})\| \\
&< \|\Sigma^{-\frac{1}{2}}Q(x_\mathsf{A}^{\mp} - x_\mathsf{A})\|
\end{aligned}$$

contradicting our assumption that $x_\mathsf{A}^{\mp}$ is optimal. Therefore $x_\mathsf{A}^{\mp}$ must satisfy $w_\mathsf{A}x_\mathsf{A}^{\mp} = -w_\mathsf{IM} \cdot x_\mathsf{IM}$. $\qquad\square$

As a result of Lemma 3, we can replace the constraint in Eq. (8) with its corresponding equality constraint without changing the optimal solution.[4] The decision subject's best-response program from Lemma 1 is therefore equivalent to

$$\min_{x_\mathsf{A}' \in \mathcal{X}_A^*} \|(\Lambda^{-\frac{1}{2}}Q(x_\mathsf{A} - x_\mathsf{A}'))\|_2 \tag{10}$$

$$\text{s.t.} \quad w_\mathsf{A} \cdot x_\mathsf{A}' - (-w_\mathsf{IM} \cdot x_\mathsf{IM}) = 0 \tag{11}$$

The following lemma gives us a closed-form solution for the above optimization problem:

**Lemma 4.** *The optimal solution to the optimization problem defined in Eq. (10) and Eq. (11)*

*has the following closed form:*

$$x_\mathsf{A}^{\mp} = x_\mathsf{A} - \frac{w^\mathsf{T}x}{w_\mathsf{A}^\mathsf{T}Sw_\mathsf{A}}Sw_\mathsf{A}.$$

*Proof.* Notice that the above program has the form

$$\min_{x_\mathsf{A}' \in x_\mathsf{A}^*} \|Ax_\mathsf{A}' - b\|_2$$

$$\text{s.t.} \quad Cx_\mathsf{A}' = d$$

where $A = \Lambda^{-\frac{1}{2}}Q$, $b = \Lambda^{-\frac{1}{2}}Qx_\mathsf{A}$, $C = w_\mathsf{A}^\mathsf{T}$, and $d = -w_\mathsf{IM}^\mathsf{T}x_\mathsf{IM}$. Note the following useful equalities:

$$\begin{aligned}
A^\mathsf{T}A &= (\Lambda^{-\frac{1}{2}}Q)^\mathsf{T}\Lambda^{-\frac{1}{2}}Q = S^{-1} \\
(A^\mathsf{T}A)^{-1} &= S \\
A^\mathsf{T}b &= (\Lambda^{-\frac{1}{2}}Q)^\mathsf{T}\Lambda^{-\frac{1}{2}}Qx_\mathsf{A} = S^{-1}x_\mathsf{A}
\end{aligned}$$

---

[4]A similar argument was made by Haghtalab et al. (2020) but here we provide a proof for a more general case, where the objective function is to minimize a weighted norm instead of simply $\|x_\mathsf{A} - x_\mathsf{A}'\|_2$.

The above is a norm minimization problem with equality constraints, whose optimum $x_{\mathsf{A}}{}^{\mp}$ has the following closed form (Boyd & Vandenberghe, 2004):

$$
\begin{aligned}
x_{\mathsf{A}}{}^{\mp} &= (A^{\mathsf{T}}A)^{-1}\left(A^{\mathsf{T}}b - C^{\mathsf{T}}(C(A^{\mathsf{T}}A)^{-1}C^{\mathsf{T}})^{-1}(C(A^{\mathsf{T}}A)^{-1}A^{\mathsf{T}}b - d)\right) \\
&= S\left(S^{-1}x_{\mathsf{A}} - w_{\mathsf{A}}(w_{\mathsf{A}}{}^{\mathsf{T}}Sw_{\mathsf{A}})^{-1}(w_{\mathsf{A}}{}^{\mathsf{T}}S(S^{-1}x_{\mathsf{A}}) - (-w_{\mathsf{IM}}{}^{\mathsf{T}}x_{\mathsf{IM}}))\right) \\
&= x_{\mathsf{A}} - S\left(w_{\mathsf{A}}(w_{\mathsf{A}}{}^{\mathsf{T}}Sw_{\mathsf{A}})^{-1}(w_{\mathsf{A}}{}^{\mathsf{T}}x_{\mathsf{A}} + w_{\mathsf{IM}}{}^{\mathsf{T}}x_{\mathsf{IM}})\right) \\
&= x_{\mathsf{A}} - \frac{w^{\mathsf{T}}x}{w_{\mathsf{A}}{}^{\mathsf{T}}Sw_{\mathsf{A}}}Sw_{\mathsf{A}}
\end{aligned}
$$

$\square$

We can now compute the cost incurred by an individual with features $x$ who plays their best response $x^{\mp}$:

$$
\begin{aligned}
c(x, x^{\mp}) &= \sqrt{(x_{\mathsf{A}} - x_{\mathsf{A}}{}^{\mp})^{\mathsf{T}}S^{-1}(x_{\mathsf{A}} - x_{\mathsf{A}}{}^{\mp})} \\
&= \sqrt{\left(\frac{w^{\mathsf{T}}x}{w_{\mathsf{A}}{}^{\mathsf{T}}Sw_{\mathsf{A}}}Sw_{\mathsf{A}}\right)^{\mathsf{T}}S^{-1}\left(\frac{w^{\mathsf{T}}x}{w_{\mathsf{A}}{}^{\mathsf{T}}Sw_{\mathsf{A}}}Sw_{\mathsf{A}}\right)} \\
&= \frac{|w^{\mathsf{T}}x|}{\sqrt{w_{\mathsf{A}}{}^{\mathsf{T}}Sw_{\mathsf{A}}}}
\end{aligned}
$$

Hence an decision subject who was classified as $-1$ with feature vector $x$ has the unconstrained best response

$$
\Delta(x) = \begin{cases} x, & \text{if } \frac{|w^{\mathsf{T}}x|}{\sqrt{w_{\mathsf{A}}{}^{\mathsf{T}}Sw_{\mathsf{A}}}} \geq 2 \\ \left[x_{\mathsf{A}} - \frac{w^{\mathsf{T}}x}{w_{\mathsf{A}}{}^{\mathsf{T}}Sw_{\mathsf{A}}}Sw_{\mathsf{A}} \mid x_{\mathsf{IM}}\right], & \text{otherwise} \end{cases}
$$

which completes the proof of Theorem 1.

## C  Proofs of Propositions in Section 3.3

**Notation.**  We make use of the following additional notation:

- $v^{(i)}$ denotes the $i$-th element of a vector $v$
- For any $\mathsf{F} \in \{\mathsf{A}, \mathsf{I}, \mathsf{M}\}$, $\Delta^{\mathsf{F}} \in \mathbb{R}^{d_{\mathsf{F}}}$ denotes the vector containing only features of type $\mathsf{F}$ within the best response $\Delta(x)$.
- $\mathbf{0}$ denotes the vector whose elements are all $0$
- $A \succ B$ indicates that matrix $A - B$ is positive definite
- $e_i$ denotes the vector containing $1$ in its $i$-th component and $0$ elsewhere

### C.1  Proof of Proposition 1

*Proof.* Let $w_{\mathsf{M}}^{(m)} \neq 0$, and consider an decision subject with original features $x$ who was classified as $-1$. By Theorem 1, the actionable sub-vector of $x$'s unconstrained best response is

$$
\Delta^{\mathsf{A}}(x) = \frac{w^{\mathsf{T}}x}{w_{\mathsf{A}}{}^{\mathsf{T}}Sw_{\mathsf{A}}}S \cdot w_{\mathsf{A}} = \frac{w^{\mathsf{T}}x}{w_{\mathsf{A}}{}^{\mathsf{T}}Sw_{\mathsf{A}}}\begin{bmatrix} S_{\mathsf{I}} & 0 \\ 0 & S_{\mathsf{M}} \end{bmatrix}\begin{bmatrix} w_{\mathsf{I}} \\ w_{\mathsf{M}} \end{bmatrix} = \frac{w^{\mathsf{T}}x}{w_{\mathsf{A}}{}^{\mathsf{T}}Sw_{\mathsf{A}}}\begin{bmatrix} S_{\mathsf{I}} \cdot w_{\mathsf{I}} \\ S_{\mathsf{M}} \cdot w_{\mathsf{M}} \end{bmatrix}
$$

And in particular,

$$\Delta^{\mathsf{M}}(x) = \frac{w^{\mathsf{T}}x}{w_{\mathsf{A}}{}^{\mathsf{T}}Sw_{\mathsf{A}}} S_{\mathsf{M}} \cdot w_{\mathsf{M}}$$

Since $x$ was initially classified as $-1$, we have $w^{\mathsf{T}}x < 0$, which means $\frac{w^{\mathsf{T}}x}{w_{\mathsf{A}}Sw_{\mathsf{A}}} \neq 0$. For convenience, let $c = \frac{w^{\mathsf{T}}x}{w_{\mathsf{A}}Sw_{\mathsf{A}}}$. We have

$$\Delta^{\mathsf{M}}(x) - x_{\mathsf{M}} = cS_{\mathsf{M}}w_{\mathsf{M}} - x_{\mathsf{M}} = S_{\mathsf{M}}(cw_{\mathsf{M}} - S_{\mathsf{M}}{}^{-1}x_{\mathsf{M}})$$

Now examine the following:

$$(cw_{\mathsf{M}} - S_{\mathsf{M}}{}^{-1}x_{\mathsf{M}})^{(m)} = cw_{\mathsf{M}}^{(m)} - (S_{\mathsf{M}}^{-1}x_{\mathsf{M}})^{(m)}$$

$$= cw_{\mathsf{M}}^{(m)} - \sum_{i=1}^{d_{\mathsf{M}}}(S_{\mathsf{M}}^{-1})^{(im)}x_{\mathsf{M}}{}^{(m)}$$

Recall that $cw_{\mathsf{M}}^{(m)} \neq 0$. Hence if $\sum_{i=1}^{d_{\mathsf{M}}}(S_{\mathsf{M}}^{-1})^{(im)} = 0$, or if

$$x_{\mathsf{M}}^{(m)} \neq \frac{cw_{\mathsf{M}}^{(m)}}{\sum_{i=1}^{d_{\mathsf{M}}}(S_{\mathsf{M}}^{-1})^{(im)}},$$

then $(cw_{\mathsf{M}} - S_{\mathsf{M}}{}^{-1}x_{\mathsf{M}})^{(m)} \neq 0$, and therefore $cw_{\mathsf{M}} - S_{\mathsf{M}}^{-1}x_{\mathsf{M}} \neq \mathbf{0}$. Since $S_{\mathsf{M}}$ is positive definite, it has full rank, which implies

$$\Delta^{\mathsf{M}}(x) - x_{\mathsf{M}} = S_{\mathsf{M}}(cw_{\mathsf{M}} - S_{\mathsf{M}}^{-1}x_{\mathsf{M}}) \neq \mathbf{0}$$

as required. With this, we have shown that when there exists a manipulated feature $x^{(m)}$ whose corresponding coefficient $w_{\mathsf{A}}{}^{(m)} \neq 0$, the classifier is vulnerable to changes in the manipulated features by the vast majority of decision subjects. $\square$

### C.2 Proof of Proposition 2

*Proof.* Consider a decision subject with features $x$ such that $h(x) = -1$. Suppose $x$ can flip this classification result by performing the improving best response $\Delta_{\mathsf{I}}(x)$, which implies that the cost of that action is no greater than 2 for this decision subject. We therefore have:

$$2 \geq c(x, \Delta_{\mathsf{I}}(x)) = \frac{|w^{\mathsf{T}}x|}{\sqrt{w_{\mathsf{I}}{}^{\mathsf{T}}S_{\mathsf{I}}w_{\mathsf{I}}}} > \frac{|w^{\mathsf{T}}x|}{\sqrt{w_{\mathsf{I}}{}^{\mathsf{T}}S_{\mathsf{I}}w_{\mathsf{I}} + w_{\mathsf{M}}{}^{\mathsf{T}}S_{\mathsf{M}}w_{\mathsf{M}}}} = \frac{|w^{\mathsf{T}}x|}{\sqrt{w_{\mathsf{A}}{}^{\mathsf{T}}Sw_{\mathsf{A}}}} = c(x, \Delta(x))$$

where the strict inequality is due to the fact that $S_{\mathsf{M}} \succ 0$ and $w_{\mathsf{M}} \neq \mathbf{0}$. As we have shown that $c(x, \Delta(x)) < 2$, we conclude whenever an decision subject can successfully flip her decision by the improving best response, she can also achieve it by performing the unconstrained best response.

On the other hand, consider the case when the unconstrained best response of a decision subject with features $x^*$ has cost exactly 2:

$$2 = c(x^*, \Delta(x^*)) = \frac{|w^{\mathsf{T}}x^*|}{\sqrt{w_{\mathsf{A}}{}^{\mathsf{T}}Sw_{\mathsf{A}}}} = \frac{|w^{\mathsf{T}}x^*|}{\sqrt{w_{\mathsf{I}}{}^{\mathsf{T}}S_{\mathsf{I}}w_{\mathsf{I}} + w_{\mathsf{M}}{}^{\mathsf{T}}S_{\mathsf{M}}w_{\mathsf{M}}}} < \frac{|w^{\mathsf{T}}x^*|}{\sqrt{w_{\mathsf{I}}{}^{\mathsf{T}}S_{\mathsf{I}}w_{\mathsf{I}}}} = c(x^*, \Delta_{\mathsf{I}}(x^*))$$

where the strict inequality is due to the fact that $S_{\mathsf{M}} \succ 0$ and $w_{\mathsf{M}} \neq \mathbf{0}$. As we have shown that $c(x^*, \Delta_{\mathsf{I}}(x^*)) > 2$, we conclude that while the unconstrained best response is viable for this decision subject, the improving best response is not. $\square$

### C.3 Proof of Proposition 3

**Proposition 7** (Correlations between Features May Reduce Cost). *For any cost matrix $S^{-1}$ and any nontrivial classifier $h$, there exist indices $k, \ell \in [d_\mathsf{A}]$ and $\tau \in \mathbb{R}$ such that every feature vector $x$ has lower best-response cost under the cost matrix $\tilde{S}^{-1}$ given by*

$$\tilde{S}_{ij}^{-1} = \tilde{S}_{ji}^{-1} = \begin{cases} S_{ij}^{-1} + \tau, & \text{if } i = k, j = \ell \\ S_{ij}^{-1}, & \text{otherwise} \end{cases}$$

*than under $S^{-1}$; that is, $c_{\tilde{S}^{-1}}(x, \Delta(x)) < c_{S^{-1}}(x, \Delta(x))$ for all $x$.*

*Proof.* Consider any cost matrix $S^{-1} \in \mathbb{R}^{d_\mathsf{A} \times d_\mathsf{A}}$ and any nontrivial classifier $h$ (i.e. $h$ does not assign every $x$ the same prediction). Since $S^{-1}$ is positive definite, so is its inverse $S$, and all of their diagonal entries are positive. And since $h$ is nontrivial, it must contain a nonzero coefficient $w_i \neq 0$. Additionally, let $w_j$ be any other coefficient.

Let $\tilde{S}^{-1} = S^{-1} + \tau(e_i e_j^\mathsf{T} + e_j e_i^\mathsf{T})$ for some constant $\tau \in \mathbb{R}$ to be set later. We claim that there exists $\tau$ such that the best-response adaptation always costs less under $\tilde{S}^{-1}$ than $S^{-1}$. To do so, we compute the inverse of $\tilde{S}^{-1}$ and invoke the closed-form cost expression given by Theorem 1.

To begin computing the inverse, note that by the Sherman-Morrison-Woodbury formula (Golub & Van Loan, 2013),

$$\tilde{S} = \left(\tilde{S}^{-1}\right)^{-1} = S - \tau S \begin{bmatrix} e_i & e_j \end{bmatrix} \left(I + \tau \begin{bmatrix} e_j^\mathsf{T} \\ e_i^\mathsf{T} \end{bmatrix} S \begin{bmatrix} e_i & e_j \end{bmatrix}\right)^{-1} \begin{bmatrix} e_j^\mathsf{T} \\ e_i^\mathsf{T} \end{bmatrix} S \tag{12}$$

$$= S - \tau S \begin{bmatrix} e_i & e_j \end{bmatrix} \left(I + \tau \begin{bmatrix} S_{ij} & S_{jj} \\ S_{ii} & S_{ij} \end{bmatrix}\right)^{-1} \begin{bmatrix} e_j^\mathsf{T} \\ e_i^\mathsf{T} \end{bmatrix} S \tag{13}$$

$$= S - \tau S \begin{bmatrix} e_i & e_j \end{bmatrix} \left[\tau\left(\frac{1}{\tau}I + \begin{bmatrix} S_{ij} & S_{jj} \\ S_{ii} & S_{ij} \end{bmatrix}\right)\right]^{-1} \begin{bmatrix} e_j^\mathsf{T} \\ e_i^\mathsf{T} \end{bmatrix} S \tag{14}$$

$$= S - \tau S \begin{bmatrix} e_i & e_j \end{bmatrix} \tau^{-1} \begin{bmatrix} \frac{1}{\tau} + S_{ij} & S_{jj} \\ S_{ii} & \frac{1}{\tau} + S_{ij} \end{bmatrix}^{-1} \begin{bmatrix} e_j^\mathsf{T} \\ e_i^\mathsf{T} \end{bmatrix} S \tag{15}$$

$$= S - S \begin{bmatrix} e_i & e_j \end{bmatrix} \underbrace{\begin{bmatrix} \frac{1}{\tau} + S_{ij} & S_{jj} \\ S_{ii} & \frac{1}{\tau} + S_{ij} \end{bmatrix}^{-1}}_{T} \begin{bmatrix} e_j^\mathsf{T} \\ e_i^\mathsf{T} \end{bmatrix} S \tag{16}$$

Clearly, we can ensure that $T$ is invertible by setting $\tau$ so that $\det(T) \neq 0$. But as the following lemmas show, we can actually say much more: $\det(T)$ can be made either positive or negative, and moreover, both can be accomplished with a choice of $\tau > 0$ or $\tau < 0$. This flexibility in choosing $\tau$ will become crucial later.

First, we need the following useful fact about positive definite matrices:

**Lemma 5** (Off-diagonal entries of a positive definite matrix). *If $A \in \mathbb{R}^{n \times n}$ is symmetric positive definite, then for all $i, j \in [n]$, $\sqrt{A_{ii}A_{jj}} > |A_{ij}|$.*

*Proof.* By positive definiteness, we have, for any nonzero $\alpha, \beta \in \mathbb{R}$,

$$(\alpha e_i + \beta e_j)^\mathsf{T} A(\alpha e_i + \beta e_j) = \alpha^2 A_{ii} + \beta^2 A_{jj} + 2\alpha\beta A_{ij} > 0$$

For a choice of $\alpha = -A_{ij}$ and $\beta = A_{ii}$, we have

$$A_{ij}^2 A_{ii} + A_{ii}^2 A_{jj} - 2A_{ij}^2 A_{ii} = A_{ii}(A_{ii}A_{jj} - A_{ij}^2) > 0$$

Since $A_{ii} > 0$, we must have $A_{ii}A_{jj} - A_{ij}^2 > 0$, from which the claim follows. $\qquad\square$

Now we can characterize the possible settings of $\tau$ and $\det(T)$:

**Lemma 6** (Possible settings of $\tau$)**.** *There exist $\tau_{\max}, \tau_{\min} > 0$ such that the following hold:*

*1.* $\det(T) > 0$ *for any $\tau \in \mathbb{R}$ such that $\tau_{\max} \geq |\tau| > 0$.*

*2.* $\det(T) < 0$ *for any $\tau \in \mathbb{R}$ such that $\tau_{\min} \leq |\tau|$.*

*Proof.* To prove the first claim, note that having

$$\det(T) = \left( \frac{1}{\tau} + S_{ij} \right)^2 - S_{ii} S_{jj} > 0$$

is equivalent to

$$\left| \frac{1}{\tau} + S_{ij} \right| > \sqrt{S_{ii} S_{jj}}$$

It suffices to choose $\tau$ such that

$$\left| \frac{1}{\tau} \right| - |S_{ij}| > \sqrt{S_{ii} S_{jj}}$$

$$\frac{1}{|\tau|} > \sqrt{S_{ii} S_{jj}} + |S_{ij}|$$

So any $\tau$ such that $0 < |\tau| < \left( \sqrt{S_{ii} S_{jj}} + |S_{ij}| \right)^{-1}$ results in $\det(T) > 0$. Analogously, for the second claim, a sufficient condition for $\det(T) < 0$ is that

$$\frac{1}{|\tau|} < \sqrt{S_{ii} S_{jj}} - |S_{ij}|$$

By Lemma 5, the right-hand side is positive. Hence it suffices to pick any $\tau$ such that

$$|\tau| > \left( \sqrt{S_{ii} S_{jj}} - |S_{ij}| \right)^{-1}.$$

$\square$

With this lemma in place, we can describe the difference between the inverses of $S^{-1}$ and $\tilde{S}^{-1}$. Denote this matrix by $E = S - \tilde{S}$. We show the following:

**Lemma 7** (Difference between inverse cost matrices)**.** *The $k, \ell$-th entry of $E$ has the following form:*

$$E_{k\ell} = \frac{1}{\det(T)} \left( E'_{k\ell} + \frac{1}{\tau} E''_{k\ell} \right)$$

*where $E'_{k\ell}$ and $E''_{k\ell}$ do not depend on $\tau$.*

*Proof.* Assume that $\tau$ has been chosen so that $\det(T) \neq 0$, as Lemma 6 showed to be possible. We then have

$$T^{-1} = \frac{1}{\det(T)} \begin{bmatrix} \frac{1}{\tau} + S_{ij} & -S_{jj} \\ -S_{ii} & \frac{1}{\tau} + S_{ij} \end{bmatrix}$$

Thus continuing from equation 16, we have

$$\tilde{S} = S - \frac{1}{\det(T)} \underbrace{S \begin{bmatrix} e_i & e_j \end{bmatrix} \begin{bmatrix} \frac{1}{\tau} + S_{ij} & -S_{jj} \\ -S_{ii} & \frac{1}{\tau} + S_{ij} \end{bmatrix} \begin{bmatrix} e_j^\mathsf{T} \\ e_i^\mathsf{T} \end{bmatrix} S}_{V}$$

It can be verified that $V$ is a $d_{\mathsf{A}} \times d_{\mathsf{A}}$ matrix whose only nonzero entries are

$$V_{ii} = -S_{jj}, \qquad V_{jj} = -S_{ii}, \qquad V_{ij} = V_{ji} = \frac{1}{\tau} + S_{ij}$$

Next we evaluate the $d_{\mathsf{A}} \times d_{\mathsf{A}}$ matrix $SVS$. For any $k, \ell \in [d_{\mathsf{A}}]$, we have

$$
\begin{aligned}
(SVS)_{k\ell} &= \sum_{i'=1}^{d_{\mathsf{A}}} \sum_{j'=1}^{d_{\mathsf{A}}} S_{ki'} V_{i'j'} S_{j'\ell} \\
&= S_{ki} V_{ii} S_{i\ell} + S_{ki} V_{ij} S_{j\ell} + S_{kj} V_{ji} S_{i\ell} + S_{kj} V_{jj} S_{j\ell} && (V \text{ has four nonzero entries}) \\
&= V_{ii} S_{ki} S_{i\ell} + V_{jj} S_{kj} S_{j\ell} + V_{ij} (S_{ki} S_{j\ell} + S_{kj} S_{i\ell}) && (V_{ij} = V_{ji}) \\
&= -S_{jj} S_{ki} S_{i\ell} - S_{ii} S_{kj} S_{j\ell} + \left( \frac{1}{\tau} + S_{ij} \right) (S_{ki} S_{j\ell} + S_{kj} S_{i\ell}) \\
&= \underbrace{-S_{jj} S_{ki} S_{i\ell} - S_{ii} S_{kj} S_{j\ell} + S_{ij}(S_{ki} S_{j\ell} + S_{kj} S_{i\ell})}_{E'_{k\ell}} + \frac{1}{\tau} \underbrace{(S_{ki} S_{j\ell} + S_{kj} S_{i\ell})}_{E''_{k\ell}}
\end{aligned}
$$

which proves the claim. $\qquad\square$

We now compute the marginal best-response cost incurred due to the difference between the inverse cost matrices, $E = S - \tilde{S}$. We have

$$
\begin{aligned}
w_{\mathsf{A}}^{\mathsf{T}} E w_{\mathsf{A}} &= \sum_{k=1}^{d_{\mathsf{A}}} \sum_{\ell=1}^{d_{\mathsf{A}}} w_k w_\ell E_{k\ell} \\
&= \frac{1}{\det(T)} \sum_{k=1}^{d_{\mathsf{A}}} \sum_{\ell=1}^{d_{\mathsf{A}}} w_k w_\ell \left( E'_{k\ell} + \frac{1}{\tau} E''_{k\ell} \right) && (\text{by Lemma 7}) \\
&= \frac{1}{\det(T)} \left[ \underbrace{\sum_{k=1}^{d_{\mathsf{A}}} \sum_{\ell=1}^{d_{\mathsf{A}}} w_k w_\ell E'_{k\ell}}_{E'} + \frac{1}{\tau} \underbrace{\sum_{k=1}^{d_{\mathsf{A}}} \sum_{\ell=1}^{d_{\mathsf{A}}} w_k w_\ell E''_{k\ell}}_{E''} \right]
\end{aligned}
$$

By Lemma 6, there exists $\tau \neq 0$ such that

$$\operatorname{sign}(\det(T)) = -\operatorname{sign}(E') \quad \text{and} \quad \operatorname{sign}(\tau) = -\operatorname{sign}(\det(T)) \cdot \operatorname{sign}(E'')$$

Such a choice of $\tau$ results in $w_{\mathsf{A}}^{\mathsf{T}} E w_{\mathsf{A}} < 0$. Finally by Theorem 1, we have for all $x$ that

$$c_{\tilde{S}^{-1}}(x, \Delta_{\tilde{S}^{-1}}(x)) = \frac{|w^{\mathsf{T}} x|}{\sqrt{w_{\mathsf{A}}^{\mathsf{T}} \tilde{S} w_{\mathsf{A}}}} = \frac{|w^{\mathsf{T}} x|}{\sqrt{w_{\mathsf{A}}^{\mathsf{T}} S w_{\mathsf{A}} - w_{\mathsf{A}}^{\mathsf{T}} E w_{\mathsf{A}}}} < \frac{|w^{\mathsf{T}} x|}{\sqrt{w_{\mathsf{A}}^{\mathsf{T}} S w_{\mathsf{A}}}} = c_{S^{-1}}(x, \Delta_{S^{-1}}(x))$$

which completes the proof. $\qquad\square$

### C.4 Proof of Proposition 4

*Proof.* Let the cost covariance matrices for groups $\Phi$ and $\Psi$ be

$$S_{\Psi}^{-1} = \begin{bmatrix} S_{\mathsf{I}}^{-1} & 0 \\ 0 & S_{\mathsf{M},\Phi}^{-1} \end{bmatrix}, \qquad S_{\Phi}^{-1} = \begin{bmatrix} S_{\mathsf{I}}^{-1} & 0 \\ 0 & S_{\mathsf{M},\Psi}^{-1} \end{bmatrix}$$

Here, we see that both groups have the same cost of changing improvable features, as represented in the cost submatrix $S_{\mathsf{I}}^{-1}$. However, the cost of manipulation for group $\Phi$ is higher than that of group $\Psi$, namely $S_{\mathsf{M},\Phi}^{-1} \succ S_{\mathsf{M},\Psi}^{-1}$.

We are now equipped to compare the costs for the two decision subjects:

$$c(x_\phi, \Delta(x_\phi)) = \frac{|w^\mathsf{T} x_\phi|}{\sqrt{w_\mathsf{A}{}^\mathsf{T} S_\Phi w_\mathsf{A}}} = \frac{|w^\mathsf{T} x|}{\sqrt{w_\mathsf{I}{}^\mathsf{T} S_\mathsf{I} w_\mathsf{I} + w_\mathsf{M}{}^\mathsf{T} \cdot S_{\mathsf{M},\Phi} \cdot w_\mathsf{M}}}$$

$$c(x_\psi, \Delta(x_\psi)) = \frac{|w^\mathsf{T} x_\psi|}{\sqrt{w_\mathsf{A}{}^\mathsf{T} S_\Psi w_\mathsf{A}}} = \frac{|w^\mathsf{T} x|}{\sqrt{w_\mathsf{I}{}^\mathsf{T} S_\mathsf{I} w_\mathsf{I} + w_\mathsf{M}{}^\mathsf{T} \cdot S_{\mathsf{M},\Psi} \cdot w_\mathsf{M}}}$$

Since $S_{\mathsf{M},\Phi}^{-1} \succ S_{\mathsf{M},\Psi}^{-1}$, we have $S_{\mathsf{M},\Phi} \prec S_{\mathsf{M},\Psi}$. And since $w_\mathsf{M} \neq \mathbf{0}$, this implies $0 < w_\mathsf{M}{}^\mathsf{T} S_{\mathsf{M},\Phi} w_\mathsf{M} < w_\mathsf{M}{}^\mathsf{T} \cdot S_{\mathsf{M},\Psi} \cdot w_\mathsf{M}$. As a result, $c(x_\phi, \Delta(x_\phi)) > c(x_\psi, \Delta(x_\psi))$ as required. $\qquad\square$

## D  Proofs and Derivations in Section 4

### D.1  Proof of Proposition 5

*Proof.* We want to show that the standard strategic risk conditioned on an unchanged true label is upper-bounded by the first term in our model designer's objective, $R_\mathsf{M}(h)$:

$$\mathbb{E}_{x \sim \mathcal{D}} [\mathbb{1}[h(x_*) \neq y] \mid \Delta(y) = y] \leq \mathbb{E}_{x \sim \mathcal{D}} [\mathbb{1}(h(x_*^\mathsf{M}) \neq y)]$$

We assume that the manipulating best response is more likely to result in a positive prediction than the unconstrained best response, given that the true labels do not change:

$$\mathbb{E}_{x \sim \mathcal{D}} [\mathbb{1}[h(x_*) \neq y] \mid \Delta(y) = y] \leq \mathbb{E}_{\mathcal{D}} [\mathbb{1}[h(x_*^\mathsf{M}) \neq y] \mid \Delta_\mathsf{M}(y) = y] \tag{17}$$

We therefore have:

$$\begin{aligned}
&\mathbb{E}_{x \sim \mathcal{D}} [\mathbb{1}(h(x_*^\mathsf{M}) \neq y)] \\
&= \mathbb{E}_{x \sim \mathcal{D}} [\mathbb{1}(h(x_*^\mathsf{M}) \neq y) \mid \Delta_\mathsf{M}(y) \neq y] \cdot \Pr[\Delta_\mathsf{M}(y) \neq y] \\
&\qquad + \mathbb{E}_{x \sim \mathcal{D}} [\mathbb{1}(h(x_*^\mathsf{M}) \neq y) \mid \Delta_\mathsf{M}(y) = y] \cdot \Pr[\Delta_\mathsf{M}(y) = y] \\
&= \mathbb{E}_{x \sim \mathcal{D}} [\mathbb{1}(h(x_*^\mathsf{M}) \neq y) \mid \Delta_\mathsf{M}(y) = y] && (\Pr[\Delta_\mathsf{M}(y) = y] = 1) \\
&\geq \mathbb{E}_{x \sim \mathcal{D}} [\mathbb{1}(h(x_*) \neq y) \mid \Delta(y) = y] && \text{(by equation 17)}
\end{aligned}$$

$$\square$$

### D.2  Proof of Proposition 6

*Proof.* Let $\mathcal{D}^*$ be the distribution induced by deploying classifier $h$. By the covariate shift assumption, $\Pr_{\mathcal{D}^*}(Y = y | X = x) = \Pr_\mathcal{D}(Y = y | X = x)$. Therefore

$$\begin{aligned}
\Pr_{x \sim \mathcal{D}^*} [y(x) = +1] &= \mathbb{E}_{\mathcal{D}^*} [\mathbb{1}[y(x) = +1]] \\
&= \int \mathbb{1}[y(x) = +1] \Pr_{\mathcal{D}^*}(X = x) dx \\
&= \int \mathbb{1}[y(x) = +1] \frac{\Pr_{\mathcal{D}^*}(X = x)}{\Pr_D(X = x)} \Pr_\mathcal{D}(X = x) dx \\
&= \int \mathbb{1}[y(x) = +1] \omega_h(x) \Pr_\mathcal{D}(X = x) dx \\
&= \mathbb{E}_{\mathcal{D}} [\omega_h(x) \mathbb{1}[y(x) = +1]]
\end{aligned}$$

This implies

$$\Pr_{x\sim\mathcal{D}^*}[y(x) = +1] \geq \Pr_{x\sim\mathcal{D}}[y(x) = +1] \iff \mathbb{E}_{\mathcal{D}}\left[(\omega_h(x) - 1)\mathbb{1}[y(x) = +1]\right] \geq 0 \tag{18}$$

By similar reasoning, we have

$$\Pr_{x\sim\mathcal{D}^*}[h(x) = +1] = \mathbb{E}_{\mathcal{D}^*}\left[\mathbb{1}[h(x) = +1]\right] = \mathbb{E}_{\mathcal{D}}\left[\omega_h(x)\mathbb{1}[h(x) = +1]\right]$$

which implies

$$\Pr_{x\sim\mathcal{D}^*}[h(x) = +1] \geq \Pr_{x\sim\mathcal{D}}[h(x) = +1] \iff \mathbb{E}_{\mathcal{D}}\left[(\omega_h(x) - 1)\mathbb{1}[h(x) = +1]\right] \geq 0 \tag{19}$$

It is easy to verify that $\mathbb{E}_{x\sim\mathcal{D}}[\omega_h(x)] = 1$, and this gives us

$$\mathbb{E}_{\mathcal{D}}\left[(\omega_h(x) - 1)\mathbb{1}[y(x) = +1]\right] = \mathrm{Cov}_{\mathcal{D}}(\omega_h(x), \mathbb{1}[y(x) = +1]) \tag{20}$$

$$\mathbb{E}_{\mathcal{D}}\left[(\omega_h(x) - 1)\mathbb{1}[h(x) = +1]\right] = \mathrm{Cov}_{\mathcal{D}}(\omega_h(x), \mathbb{1}[h(x) = +1]) \tag{21}$$

By equation 18, equation 19, and equation 20, the condition

$$\Pr_{x\sim\mathcal{D}^*}[h(x) = +1] \geq \Pr_{x\sim\mathcal{D}}[h(x) = +1] \iff \Pr_{x\sim\mathcal{D}^*}[y(x) = +1] \geq \Pr_{x\sim\mathcal{D}}[y(x) = +1]$$

is equivalent to the condition

$$\mathrm{Cov}_{\mathcal{D}}(\omega_h(x), \mathbb{1}[y(x) = +1]) \geq 0 \iff \mathrm{Cov}_{\mathcal{D}}(\omega_h(x), \mathbb{1}[h(x) = +1]) \geq 0$$

$\square$

### D.3 Derivations for the model designer's objective function

Now that we have obtained a closed-form expression for both the unconstrained and improving best response from the decision subjects, we can analyze the objective function for the model designer and the model that would be deployed at equilibrium. Recall that the objective function for the model designer is

$$\min_{w\in\mathbb{R}^{d+1}} \mathbb{E}_{x\sim\mathcal{D}}\left[\mathbb{1}(h(\Delta_{\mathsf{M}}(x)) \neq y)\right] + \lambda \mathbb{E}_{x\sim\mathcal{D}}\left[\mathbb{1}(h(\Delta_{\mathsf{I}}(x)) \neq +1)\right]$$

By Theorem 1, $h(\Delta_{\mathsf{M}}(x))$ has the closed form

$$h(\Delta_{\mathsf{M}}(x)) = \begin{cases} +1 & \text{if } w\cdot x \geq -2\sqrt{w_{\mathsf{M}}^{\mathsf{T}}S_{\mathsf{M}}w_{\mathsf{M}}} \\ -1 & \text{otherwise} \end{cases}$$

$$= 2\cdot\mathbb{1}\left[w\cdot x \geq -2\sqrt{w_{\mathsf{M}}^{\mathsf{T}}S_{\mathsf{M}}w_{\mathsf{M}}}\right] - 1$$

and similarly,

$$h(\Delta_{\mathsf{I}}(x)) = 2\cdot\mathbb{1}\left[w\cdot x \geq -2\sqrt{w_{\mathsf{I}}^{\mathsf{T}}S_{\mathsf{I}}w_{\mathsf{I}}}\right] - 1$$

The model designer's objective can then be re-written as follows:

$$\mathbb{E}_{x\sim D}\left[\mathbb{1}[h(\Delta_{\mathsf{M}}(x)) \neq y] + \lambda\mathbb{1}[h(\Delta_{\mathsf{I}}(x)) \neq +1]\right]$$

$$= \mathbb{E}_{x\sim\mathcal{D}}\left[1 - \frac{1}{2}(1 + h(\Delta_{\mathsf{M}}(x))\cdot y) + \lambda(1 - \frac{1}{2}(1 + h(\Delta_{\mathsf{I}}(x))\cdot 1))\right]$$

$$= \mathbb{E}_{x\sim\mathcal{D}}\left[\frac{1}{2}(1 + \lambda) - \frac{1}{2}h(\Delta_{\mathsf{M}}(x))\cdot y - \frac{\lambda}{2}h(\Delta_{\mathsf{I}}(x))\right]$$

Removing the constants, the objective function becomes:

$$\min_w \mathbb{E}_{x \sim \mathcal{D}} \left[ \lambda - h(\Delta_{\mathsf{M}}(x)) \cdot y - \lambda h(\Delta_{\mathsf{I}}(x)) \right]$$

$$= \min_w \mathbb{E}_{x \sim \mathcal{D}} \left[ - \left( 2 \cdot \mathbb{1} \left[ w \cdot x \geq -2\sqrt{w_{\mathsf{M}}{}^{\mathsf{T}} S_{\mathsf{M}} w_{\mathsf{M}}} \right] - 1 \right) \cdot y(x) - 2\lambda \cdot \mathbb{1} \left[ w \cdot x \geq -2\sqrt{w_{\mathsf{I}}{}^{\mathsf{T}} S_I w_{\mathsf{I}}} \right] \right]$$

Re-organizing the above equations, we can turn the model designer's *constrained* optimization problem in equation 7 into the following *unconstrained* problem:

$$\min_{w \in \mathbb{R}^d} \mathbb{E}_{x \sim \mathcal{D}} \left[ - \left( 2 \cdot \mathbb{1} \left[ w^{\mathsf{T}} x \geq -2\sqrt{\Omega_{\mathsf{M}}} \right] - 1 \right) \cdot y - 2\lambda \cdot \mathbb{1} \left[ w^{\mathsf{T}} x \geq -2\sqrt{\Omega_{\mathsf{I}}} \right] \right] \tag{22}$$

The optimization problem in equation 22 is intractable since both the objective and the constraints are non-convex. To overcome this difficulty, we train our classifier by replacing the 0-1 loss function with a convex surrogate loss $\sigma(x) = \log \left( \frac{1}{1+e^{-x}} \right)$. This results in the following ERM problem:

$$\min_{w \in \mathbb{R}^d} \frac{1}{n} \sum_{i=1}^{n} \left[ - \sigma \left( y_i \cdot (w^{\mathsf{T}} x_i + 2\sqrt{\Omega_{\mathsf{M}}}) \right) - \lambda \cdot \sigma(w^{\mathsf{T}} x_i + 2\sqrt{\Omega_{\mathsf{I}}}) \right] \tag{23}$$

**Conditionally Actionable Features.** In practice, individuals can often only change some features in either a positive or negative direction, but not both. However, modeling this restriction on the decision subject's side precludes a closed-form solution. Instead, we strongly disincentivize such moves in the model designer's objective function. The idea is that if the model designer is punished for encouraging an illegal action, the announced classifier will not incentivize such moves from decision subjects. The result is that decision subjects encounter an *implicit* direction constraint on the relevant variables. To that end, we construct a vector $\mathsf{dir} \in \{-1, 0, +1\}^d$ where $\mathsf{dir}_i$ represents the prohibited direction of change for the corresponding feature $x_i$; that is, $\mathsf{dir}_i = +1$ if $x_i$ should not be allowed to increase, $-1$ if it should not decrease, and $0$ if there are no direction constraints. We then append the following penalty term to the model designer's objective in Eq. (7):

$$-\eta \cdot \sum_{i=1}^{d} \max(\mathsf{dir}_i \cdot (\Delta(x) - x)_i, 0) \tag{24}$$

where $\eta > 0$ is a hyperparameter representing the weight given to this penalty term. Eq. (24) penalizes the weights of partially actionable features so that decision subjects would prefer to move towards a certain direction.

## E    Additional Related Work

**Strategic Classification.** There has been extensive research on strategic behavior in classification Hardt et al. (2016a); Cai et al. (2015); Chen et al. (2018); Dong et al. (2018); Dekel et al. (2010); Chen et al. (2020). Hardt et al. (2016a) was the first to formalize strategic behavior in classification based on a sequential two-player game (i.e. a Stackelberg game) between decision subjects and classifiers. Since then, other similar Stackelberg formulations have been studied Balcan et al. (2015). Dong et al. (2018) considers the setting in which decision subjects arrive in an online fashion and the learner lacks full knowledge of decision subjects' utility functions. More recently, Chen et al. (2020) proposes a learning algorithm with non-smooth utility and loss functions that adaptively partitions the learner's action space according to the decision subject's best responses.

**Recourse.** The concept of *recourse* in machine learning was first introduced in (Ustun et al., 2019). There, an integer programming solution was developed to offer actionable recourse from a linear classifier. Our work builds on theirs by considering strategic actions from decision subjects, as well as by aiming to incentivize honest improvement. Venkatasubramanian & Alfano (2020) discusses a more adequate conceptualization and

operationalization of recourse. Karimi et al. (2020a) provides a thorough survey of algorithmic recourse in terms of its definitions, formulations, solutions, and prospects. Inspired by the concept of recourse, Dean et al. (2020) develops a reachability problem to capture the ability of models to accommodate arbitrary changes in the interests of individuals in recommender systems. Bellamy et al. (2018) builds toolkits for actionable recourse analysis. Furthermore, Gupta et al. (2019) studies how to mitigate disparities in recourse across populations.

**Causal Modeling of Features.** A flurry of recent papers have demonstrated the importance of understanding causal factors for achieving fairness in machine learning (Wang et al., 2019; Bhatt et al., 2020; Bechavod et al., 2020; Miller et al., 2020; Shavit et al., 2020). Miller et al. (2020) studies distinctions between gaming and improvement from a causal perspective. Shavit et al. (2020) provides efficient algorithms for simultaneously minimizing predictive risk and incentivizing decision subjects to improve their outcomes in a linear setting. In addition, Karimi et al. (2020b) develops methods for discovering recourse-achieving actions with high probability given limited causal knowledge. In contrast to these works, we explicitly separate improvable features from manipulated features when maximizing decision subjects' payoffs. Our work also broadly relates to the concept of *intervention* in the literature of causal inference (Eberhardt & Scheines, 2007). In our work, the actionability of a feature is always factual, meaning it is always feasible to change those features. This is closely related to the concept of *last trial* in causal inference, which refers to the interventions that one could run in the real-world (which would rule out the interventions on age) (Hernán et al., 2022).

**Incentive Design.** Like our work, Kleinberg & Raghavan (2020) discusses how to incentivize decision subjects to improve a certain subset of features. Next, Haghtalab et al. (2020) shows that an appropriate projection is an optimal linear mechanism for strategic classification, as well as an *approximate* linear threshold mechanism. Our work complements theirs by providing appropriate linear classifiers that balance accuracy and improvement. Liu et al. (2020) considers the equilibria of a dynamic decision-making process in which individuals from different demographic groups invest rationally, and compares the impact of two interventions: decoupling the decision rule by group and subsidizing the cost of investment.

**Algorithmic Fairness in Machine Learning.** Our work contributes to the broad study of algorithmic fairness in machine learning. Most common notions of group fairness include disparate impact Feldman et al. (2015), demographic parity Agarwal et al. (2018), disparate mistreatment Zafar et al. (2019), equality of opportunity Hardt et al. (2016b) and calibration Chouldechova (2017). Among them, disparities in the recourse fraction can be viewed as equality of false positive rate (FPR) in the strategic classification setting. Disparities in costs and flipsets are also relevant to counterfactual fairness Kusner et al. (2017) and individual fairness Dwork et al. (2012). Similar to our work, von Kügelgen et al. (2020) also consider the intervention cost of recourse in flipping the prediction across subgroups, investigating the fairness of recourse from a causal perspective.

### E.1  Agent's Best Response with Partially Actionable Features

Let feature $i$ represents those features that should only be non-increasing, and feature $j$ represents those features that should only be non-decreasing. Then the constraint can be represented as:

$$y_i \leq 0 \Leftrightarrow e_i^\mathsf{T} y \leq 0$$
$$y_j \geq 0 \Leftrightarrow e_j^\mathsf{T} y \geq 0$$

Assume that there are $n_-$ features that can only be changed negatively, and there are $n_+$ features that can only be changed increasingly. We can further combine those new constraints into a matrix form like $Ey \leq 0$. The other constraint can be re-written as:

$$w^\mathsf{T} y - b' \geq 0 \Leftrightarrow -w^\mathsf{T} y \leq -b',$$

therefore the optimization problem can be rewritten as:

$$\min \quad \frac{1}{2} y^{\mathsf{T}} Q y$$

$$s.t. \quad \underbrace{\begin{bmatrix} E \\ -w^{\mathsf{T}} \end{bmatrix}}_{A} y \le \underbrace{\begin{bmatrix} 0 \\ -b' \end{bmatrix}}_{b}$$

where $A$ is of the form:

$$A = \begin{bmatrix} I_{n_- \times n_-} & 0 & 0 \\ 0 & -I_{n_+ \times n_+} & 0 \\ \smile\smile & -w^{\mathsf{T}} & \smile\smile \end{bmatrix}_{(n_+ + n_- + 1) \times n}$$

# F   Additional Experimental Details and Results

In this section, we provide additional experimental results. In particular, we provide the full results with mean and standard deviation in Table 4.

## F.1   Basic information of each dataset

Table 3: Basic information of each dataset.

| Dataset | Size | Dimension | Prediction Task |
|---------|------|-----------|-----------------|
| credit | $20,000$ | 16 | To predict if a person can repay their credit card loan. |
| adult | $48,842$ | 14 | To predict whether income exceeds $50K/yr$ based on census data. |
| german | $1,000$ | 26 | To predict whether a person is good or bad credit risk. |
| spam | $4601$ | 57 | To predict if an email is a spam or not. |

## F.2   Specific Cost Matrix

We specify the cost matrix as follows:

$$S_{ij}^{-1} = \begin{cases} 1, & \text{if } i = j \text{ and } i \in \mathsf{I} \\ 0.2, & \text{if } i = j \text{ and } j \in \mathsf{M} \\ 1, & \text{if the cost of changing features } i \\ & \text{and } j \text{ are } \textit{negatively} \text{ correlated} \\ -1, & \text{if the cost of changing features } i \\ & \text{and } j \text{ are } \textit{positively} \text{ correlated} \\ 0, & \text{otherwise} \end{cases}$$

We use the credit dataset as a demonstration of how we specify the non-diagonal element in the cost matrix. For two feature variables that have a positive correlation, e.g., *CheckingAccountBalance* and *SavingsAccountBalance*, we assign $-1$ to the corresponding elements in the cost matrix $S$. For two feature variables that have a negative correlation, e.g., *CheckingAccountBalance* and *MissedPayments*, we assign $+1$ to the corresponding elements in the cost matrix.

## F.3   Computing Infrastructure

We conducted all experiments on a 3 GHz 6-Core Intel Core i5 CPU. All our methods have relatively modest computational cost and can be trained within a few minutes.

Table 4: Performance metrics for all methods over 4 real data sets with non-diagonal cost matrix. We report the mean and standard deviation for 5-fold cross validation. The constructive adaptation (CA) consistently achieves a high accuracy at deployment while providing the highest improvement rates across all four datasets.

| Dataset | Metrics | METHODS | | | |
|---------|---------|---------|---------|---------|---------|
| | | ST | DF | MP | CA |
| CREDIT | *test error* | $29.52 \pm 0.37$ | $29.66 \pm 0.40$ | $29.65 \pm 0.41$ | $29.60 \pm 0.44$ |
| | *deploy error* | $31.25 \pm 0.56$ | $29.66 \pm 0.40$ | $29.41 \pm 0.32$ | $29.49 \pm 0.38$ |
| | *improvement rate* | $46.35 \pm 3.81$ | $44.71 \pm 4.75$ | $36.76 \pm 0.53$ | $48.27 \pm 5.50$ |
| ADULT | *test error* | $23.05 \pm 0.47$ | $33.55 \pm 0.73$ | $24.94 \pm 0.52$ | $27.22 \pm 0.65$ |
| | *deploy error* | $38.64 \pm 4.46$ | $33.55 \pm 0.73$ | $26.85 \pm 0.59$ | $29.34 \pm 0.45$ |
| | *improvement rate* | $30.92 \pm 3.31$ | $60.63 \pm 29.40$ | $36.70 \pm 1.62$ | $63.79 \pm 7.80$ |
| GERMAN | *test error* | $30.85 \pm 0.82$ | $36.10 \pm 1.97$ | $33.25 \pm 1.44$ | $34.70 \pm 2.15$ |
| | *deploy error* | $33.40 \pm 1.78$ | $36.10 \pm 1.97$ | $34.60 \pm 1.94$ | $34.25 \pm 1.78$ |
| | *improvement rate* | $41.20 \pm 5.77$ | $42.10 \pm 9.07$ | $33.50 \pm 2.53$ | $56.10 \pm 6.40$ |
| SPAMBASE | *test error* | $7.11 \pm 0.52$ | $10.18 \pm 0.45$ | $11.52 \pm 0.12$ | $14.37 \pm 0.24$ |
| | *deploy error* | $22.40 \pm 3.14$ | $10.18 \pm 0.45$ | $12.92 \pm 0.58$ | $14.70 \pm 0.36$ |
| | *improvement rate* | $40.04 \pm 13.06$ | $32.46 \pm 14.63$ | $26.42 \pm 4.80$ | $43.98 \pm 6.18$ |

## F.4 Results for non-diagonal cost matrix

In real life, the specification of the cost matrix might require examining the causal correlations among different features. We consider a non-diagonal cost matrix setup based on common knowledge and describe the rationale as below. For two feature variables that have a positive correlation, e.g., *CheckingAccountBalance* and *SavingsAccountBalance*, we assign -1 to the corresponding elements in the cost matrix. For two feature variables that have a negative correlation, e.g., *CheckingAccountBalance* and *MissedPayments*, we assign +1 to the corresponding elements in the cost matrix. We also note that the non-diagonal cost matrix must be invertible under our assumption on the cost of modifying features. We provide more detailed results for each dataset in Table 4, which shows the means and standard deviations of different metrics. Compared to the empirical results of using a diagonal matrix, we achieve similar results with respect to the three evaluation criteria across all four methods.

## F.5 Additional Experimental Results for Non-Linear models

We also work with a three-layer neural network to validate the effectiveness of the oracle best response in Algorithm 1. We note that the LIME program needs to learn a local linear model for each instance, which is very time-consuming. Therefore, we downsample only 10% of data examples from the credit dataset. We follow the same setting as the linear classifier experiments. We compare our method with the static classifier in Table 5. We find out for this non-linear model setting, our approach has a higher improvement rate while preventing manipulations with the deploy error 27.72% vs. 35.64%.

Table 5: Performance metrics for non-linear models.

| Metrics | METHODS | |
|---------|---------|---------|
| | ST | CA |
| *test error* | 30.72% | 30.01% |
| *deploy error* | 35.64% | 27.72% |
| *improvement rate* | 0.99% | 2.97% |

### F.6 Flipsets

We also construct flipsets for individuals in the german dataset using the closed-form solution Eq. (4) under our trained classifier. The individual characterized as a "bad consumer" $(-1)$ is supposed to decrease their missed payments in order to flip their outcome of the classifier with respect to a non-diagonal cost matrix. In contrast, even though the individual improves their loan rate or liable individuals, the baseline classifier will still reject them. We also provide flipsets for conditionally actionable features on the credit dataset in Table 7. The individual will undesirably reduce their education level when the classifier is unaware of the partially actionable features. In contrast, the individual decreases their total overdue months instead when the direction penalty is imposed during training. [5]

Table 6: Flipset for a person denied credit by ManipulatedProof on the german dataset. The red up arrows ↑ represent increasing the values of features, while the red down arrows ↓ represent decreasing.

| Feature | Type | Original | LightTouch | ManipulatedProof |
|---|---|---|---|---|
| *LoanRateAsPercentOfIncome* | I | 3 | 3 | 2 ↓ |
| *NumberOfOtherLoansAtBank* | I | 1 | 1 | 1 |
| *NumberOfLiableIndividuals* | I | 1 | 0 ↓ | 2 ↑ |
| *CheckingAccountBalance* $\geq 0$ | I | 0 | 0 | 0 |
| *CheckingAccountBalance* $\geq 200$ | I | 0 | 0 | 0 |
| *SavingsAccountBalance* $\geq 100$ | I | 0 | 0 | 0 |
| *SavingsAccountBalance* $\geq 500$ | I | 0 | 0 | 0 |
| *MissedPayments* | I | 1 | 0 ↓ | 1 |
| *NoCurrentLoan* | I | 0 | 0 | 0 |
| *CriticalAccountOrLoansElsewhere* | I | 0 | 0 | 0 |
| *OtherLoansAtBank* | I | 0 | 0 | 0 |
| *OtherLoansAtStore* | I | 0 | 0 | 0 |
| *HasCoapplicant* | I | 0 | 0 | 0 |
| *HasGuarantor* | I | 0 | 0 | 0 |
| *Unemployed* | I | 0 | 0 | 0 |
| *LoanDuration* | M | 48 | 47 ↓ | 47 ↓ |
| *PurposeOfLoan* | M | 0 | 0 | 0 |
| *LoanAmount* | M | 4308 | 4307 ↓ | 4307 ↓ |
| *HasTelephone* | M | 0 | 0 | 0 |
| *Gender* | U | 0 | 0 | 0 |
| *ForeignWorker* | U | 0 | 0 | 0 |
| *Single* | U | 0 | 0 | 0 |
| *Age* | U | 24 | 24 | 24 |
| *YearsAtCurrentHome* | U | 4 | 4 | 4 |
| *OwnsHouse* | U | 0 | 0 | 0 |
| *RentsHouse* | U | 1 | 1 | 1 |
| *YearsAtCurrentJob* $\leq 1$ | U | 1 | 1 | 1 |
| *YearsAtCurrentJob* $\geq 4$ | U | 0 | 0 | 0 |
| *JobClassIsSkilled* | U | 1 | 1 | 1 |
| *GoodConsumer* | - | $-1$ | $+1$ ↑ | $-1$ |

---

[5]In this experiment, we implicitly assume that account balances reflect the agent's true savings.

Table 7: Flipset for an individual on Credit dataset with partially actionable features. The red up arrows ↑ represent any increasing values, while the red down arrows ↓ represent any decreasing values.

| Feature | Type | dir | Original | $\eta = 0$ | $\eta = 100$ |
|---|---|---|---|---|---|
| *EducationLevel* | I | +1 | 3 | 2 ↓ | 3 |
| *TotalOverdueCounts* | I | 0 | 1 | 1 | 1 |
| *TotalMonthsOverdue* | I | 0 | 1 | 1 | 0 ↓ |
| *MaxBillAmountOverLast6Months* | M | 0 | 0 | 0 | 0 |
| *MaxPaymentAmountOverLast6Months* | M | 0 | 0 | 0 | 0 |
| *MonthsWithZeroBalanceOverLast6Months* | M | 0 | 0 | 0 | 0 |
| *MonthsWithLowSpendingOverLast6Months* | M | 0 | 6 | 5 ↓ | 6 |
| *MonthsWithHighSpendingOverLast6Months* | M | 0 | 0 | 0 | 0 |
| *MostRecentBillAmount* | M | 0 | 0 | 0 | 0 |
| *MostRecentPaymentAmount* | M | 0 | 0 | 0 | 0 |
| *Married* | U | 0 | 1 | 1 | 1 |
| *Single* | U | 0 | 0 | 0 | 0 |
| $Age \leq 25$ | U | 0 | 0 | 0 | 0 |
| $25 \leq Age \leq 40$ | U | 0 | 0 | 0 | 0 |
| $40 \leq Age < 60$ | U | 0 | 0 | 0 | 0 |
| $Age \geq 60$ | U | 0 | 1 | 1 | 1 |
| *HistoryOfOverduePayments* | U | 0 | 1 | 1 | 1 |
| *NoDefaultNextMonth* | - | - | −1 | +1 ↑ | +1 ↑ |

