# OpenReview forum: "Learning to Incentivize Improvements from Strategic Agents "
_TMLR — Accepted by TMLR_

### Review · Reviewer_TLHt · 2023-02-07

**Summary Of Contributions:**

This paper considers the case of a model designer that needs to train a classifier in a way that will make predictions over decision subjects while these subjects themselves have the possibility to alter their features to obtain a specific prediction.
The paper assumes that decision subjects know the classifier, and the classifier knows the decision subjects’ cost function.
In other words, this is "the standard setting in strategic classification where the model designer has no strong verification power to verify truthfulness of $x^*$".
The paper explain that existing approaches in strategic classification tackle these issues by training a classifier that is robust to all adaptation. This paper aims at improving current shortcomings that can be summarized as follows: (i) the classifier from current techniques "still has suboptimal accuracy because y changes as a result of the adaptation in x" and (ii) existing approaches miss "the opportunity to encourage a profile x to truly improve to change their y".

**Audience:**

Yes

**Broader Impact Concerns:**

No broader impact concerns

**Claims And Evidence:**

Yes

**Requested Changes:**

Critical:
I have a few requests for change that I rank as critical but they might only require minor modifications and/or clarifications:
- The minimization of the constructive adaptation (CA) risk is intuitively meaningful but seems to be somewhat arbitrary. For instance, could you justify why you would not have a term (1-\lambda) in front of the manipulation risk term?
- Right before Theorem 1, the paper introduces $\hat S_F := ((S^{−1})_F)^{−1}$ but it is a bit unclear why the double inversion is necessary.
- The notations with subscripts and superscripts is not always clear. For instance, the superscript $(m)$ is used for the first time in proposition 1 both for $x$ and $\Delta$ but the meaning could be improved (in particular $\Delta^{(m)}$ does not seem to be defined).
- I don't see any proof for Proposition 3, why?

Suggestions to strengthen the work
- The introduction of actionable features $x_A = [x_I | x_M]$ seems to be defined as either a improvable or a manipulable feature. The terminology makes sense but I'm wondering whether a terminology closer to for instance the causal inference literature would make even more sense (e.g. with the concept of intervention). It could be useful to relate/explain the choice of at least the *actionable* features.
- The source code for the experiments does not seem to be given. Is that a possibility?


**Strengths And Weaknesses:**

- The paper is overall well-written
- The specific setting of the contribution is interesting and it is original as far as I can tell (however I'm not an expert in this particular area).
- The limitations are explicitly summarized in the conclusion, which provides a fair and clear view on the scope of the work.

---

> ### Author Response · Authors · 2023-02-11
> **Response to Reviewer TLHt**
>
> Thank you so much for your valuable feedback! Below we address your comments and requested changes:
>
> > ``The minimization of the constructive adaptation (CA) risk is intuitively meaningful but seems to be somewhat arbitrary. For instance, could you justify why you would not have a term $(1-\lambda)$ in front of the manipulation risk term?''
>
> Recall that the constructive adaptation risk aims to balance robustness to manipulation (namely the manipulation risk) and incentivization of improvement (namely the improvement risk), and $\lambda$ is a parameter used to capture the trade-off between these two objectives we aim to optimize for.
> It is true that there is an *implicit* $(1 - \lambda)$ term in front of the manipulation risk term that matches the $\lambda$ term in front of the improvement risk term; however, we would like to point out that having one single parameter $\lambda'$ in front of the improvement risk term is equivalent to having $\lambda$ and $(1 - \lambda)$ in front of each risk term accordingly. To see this, notice that one can divide the whole expression by $(1 - \lambda)$, or simply set $\lambda'  = \lambda/(1 - \lambda)$, and then it becomes equivalent to having both coefficients.
>
>
> > ``Right before Theorem 1, the paper introduces $\hat{S}_F = ((S^{-1})_F)^{-1}$ but it is a bit unclear why the double inversion is necessary.''
>
> This is a great question. The reason why we need to have double inversion in $\hat{S}_F = ((S^{-1})_F)^{-1}$ is because the F-submatrix of $S^{-1}$, which we denote as $(S^{-1})_F$, is not the same as the inverse of $S_F$, aka $(S_F)^{-1}$, where $S_F$ is the F-submatrix of $S$, thus $\hat{S}_F = ((S^{-1})_F)^{-1} \neq S_F$. Notice that $S^{-1}$ is PSD matrix, which implies its inverse matrix $S$ and its submatrix $S_F$ and their inverse  $S_F^{-1}$ are PSD matrix.  However, for PSD matrix, the inverse of the submatrix does not necessarily equal the submatrix's inverse.
>
>
> > ``The notations with subscripts and superscripts is not always clear. For instance, the superscript $(m)$ is used for the first time in proposition 1 both for $x$ and $\Delta$, but the meaning could be improved (in particular, $\Delta^{(m)}$ does not seem to be defined).''
>
> We apologize for the confusion. In our paper, the subscript (e.g. $x_m$) refers to the entire feature vector (e.g., $x_m\in R^{d_m}$, where $d_m$ is the total number of the manipulative features), while the superscript $(m)$ refers to the particular index of a particular manipulation feature.
> We have included a clarification in the footnote for proposition 1.
>
> > ``I don't see any proof for Proposition 3, why?''
>
> We apologize for not including the proof of Proposition 3, this is a mistake from our side; we have added it in our revision.
>
>
> > ``The introduction of actionable features seems to be defined as either a improvable or a manipulable feature. The terminology makes sense but I'm wondering whether a terminology closer to for instance the causal inference literature would make even more sense (e.g. with the concept of intervention). It could be useful to relate/explain the choice of at least the actionable features.''
>
> That is a very good question. Please let us clarify the relationship between the actionability of a feature and the concept of intervention in causal inference.
>
> In our content, actionable features refer to those that could be readily altered by human agents (e.g., their income or salary) as opposed to those that are not (e.g., their age or sex) *in real life*. Thus the actionability of a feature is always *factual*. Meanwhile, in the language of causal inference, intervention can be *counterfactual*: intervention involves randomly assigning values to a single variable and then predict what will happen to another variable. For example, in causal inference, one can intervene on the variable "age" and see its effects on the qualification $y$. In this example, age is *not* an actionable feature by our definition, because human agents can not alter their age in reality, however, it can be intervened on when one performs causal analysis to understand how it influences the predicted outcome.  Thus there is no direct mapping between the actionability of a feature and intervention causal inference terminology.
>
>
> > ``The source code for the experiments does not seem to be given. Is that a possibility?''
>
> We would like to kindly point out that the source code of our experimental result is included in the supplement section for the current submission; in our revision, we have included a link to a GitHub repository in the main paper.
>
> Please let us know if there are any further questions!

---

> > ### Comment · Reviewer_TLHt · 2023-03-20
> > **Thanks for the clarifications**
> >
> > Thanks for the clarifications.
> >
> > One minor remark: I believe you mean superscript instead of "subscription".

---

> > > ### Author Response · Authors · 2023-03-20
> > > **Re: subscript vs superscript**
> > >
> > > You are absolutely correct -- we have fixed this mistake in our revision and in the comment. Thank you for pointing it out.

---

> > ### Comment · Reviewer_5yXo · 2023-03-21
> > **Target trials**
> >
> > >  For example, in causal inference, one can intervene on the variable "age" and see its effects on the qualification
> > y. In this example, age is not an actionable feature by our definition, because human agents can not alter their age in reality, however, it can be intervened on when one performs causal analysis to understand how it influences the predicted outcome. Thus there is no direct mapping between the actionability of a feature and intervention causal inference terminology.
> >
> > I thought it worth mentioning that this view is not universal in causal inference: see for example, [Hernan & Robins](https://www.hsph.harvard.edu/miguel-hernan/wp-content/uploads/sites/1268/2023/02/hernanrobins_WhatIf_7feb23.pdf) section 3.6 (page 38) where they argue that we should only think of interventions with respect to a "target trial" that you could actually run in the world (which would rule out interventions on age). This framework seems far closer to what you have in mind with actionable features.

---

> > > ### Author Response · Authors · 2023-03-22
> > > **RE: target trails**
> > >
> > > Thank you very much for pointing out the potential relationship between our work and the concept of _target trail_ in causal inference; it does relate to our concept of _actionability_.
> > >
> > > We have added this discussion to the additional reference section (see Section E on Causal Modeling of Features in the appendix) in our revision.

---

### Review · Reviewer_EU7Q · 2023-02-19

**Summary Of Contributions:**

This paper studies the strategic classification setting. A classifier designer wishes to create a linear classifier while knowing that each data point is controlled by an agent who can change features of that data point by incurring a cost. Specifically, the paper considers features in two categories: improvable and not, with the former being broken down into two further subcategories: manipulable—i.e., features that, when changed, do not affect the true label of a data point; and improvable—features that, when changed, affect the true label of a data point. The high-level idea is that the designer wishes to discourage the agent from altering their manipulable features, but wants to encourage the agent to alter their improvable features.

The authors provide some analysis of the dynamics of this problem as a game (i.e., what is the best response function for the agent) under the assumptions of a Mahalonobis distance-based cost function for the agents.

To fit a model that encourages improvement and discourages manipulation, the authors propose minimizing a new risk function to fit the model. Because the system designer knows which features are manipulable or improvable, they encourage the model to predict the label true label if the agent manipulates but encourage the model to predict the "qualified" label if the agent improves.

They simulate the effectiveness of their model on some standard domains from the literature.

**Audience:**

No

**Claims And Evidence:**

No

**Requested Changes:**

Minor:
- References are formatted incorrectly in many places
- The handling of the causal related stuff is somewhat loose. Some example issues: social media presence as an example. As important causal factor A goes up (e.g., income), social media presence may causally increase depending on A—this is sort of the general story we have for why these manipulable factors are correlated with the label. So when causal features are improved, they are likely to change non-causal features as well.
- The best response function is not defined when it is introduced. Need a utility function for the agent. (change to h = 1 with min cost?)
- The notion of qualification is used without definition in 2.1
- “We assume certain features are known to affect the qualification” what do you mean by this, formally?
- 2.2 and 2.1 contradictory—you say in 2.1 that y always represents true qualification before adaptation, but in 2.2, you say that y can change as a result of adaptation in x
- Recourse not defined/cited

**Strengths And Weaknesses:**

Strengths:
- I don't know enough about the strategic classification space to know exactly what has been done there so far, but the authors' arguments for why the kind of model they're presenting is the right direction make sense to me.
- The idea of a new risk function that has different components focusing on different best responses is interesting.

Weaknesses:
- The model is not really handled formally and really should be. The authors claim to model the problem as a Stackelberg game but don't really achieve this, at least in the sense that they don't specify who the players are, their utility functions, the action spaces, etc. (Many of these details can be "filled in" by an informed reader, but that's not the goal here.)
- In particular, the authors never state what the utility function for the system designer is. I think this is a critical question—the central theorem for the work should relate the outcome that is achieved by the authors' methods to the utility function of the designer. We need to know "how good" this solution is relative to others if we are going to model this game theoretically.
- The paper doesn't have enough new stuff in it in my opinion. The authors' chosen setting (linear classifiers) should be tremendously theoretically tractable, so the bar here is somewhat higher—we really should be able to have theorems that relate the things we care about and not rely on empirical results.

---

> ### Author Response · Authors · 2023-02-27
> **Response to Reviewer EU7Q: Part 1**
>
> Thank you so much for your valuable feedback! Below we address your comments and requested changes:
>
>
> #### **Question 1**:
> > ``The model is not really handled formally and really should be. The authors claim to model the problem as a Stackelberg game but don't really achieve this, at least in the sense that they don't specify who the players are, their utility functions, the action spaces, etc.''
>
> We apologize for any confusion caused by the current formulation of the Stackelberg game in the Preliminaries Section (Section 2.1). We want to note that we provide more formal definitions and notation in later sections of the paper as we discuss more details, such as Section 3.2 for the definition of the utility for the agent. However, we acknowledge that when we first introduced the game in Section 2.1, we did not include all the formal definitions due to the need to provide a clear and accessible introduction to our framework.
>
> To provide further clarity, we propose including additional details:
> 1) We will specify that the game involves two players: the decision maker and strategic agents.
> 2)  The utility functions for each player are as follows: for the strategic agents, their utility for adapting their feature from $x$ to $x'$ is determined by the standard utility function in the literature of strategic classification (see, e.g., Hardt et al., 2016), which is $U(x, x') = h(x') - c(x, x')$. The decision maker's goal is to correctly classify the strategic agents based on their adapted features and corresponding new qualifications. We represent the decision maker's utility as the probability that the proposed classifier $h$ correctly predicts the label $y$ of an agent's adapted feature $\Delta(x)$. This can be expressed mathematically as $Pr(h(\Delta(x)) = y(\Delta(x)))$.
>
> 3)  The action space for each party is defined as follows: for the strategic agents, it includes all feature vectors that are within the reach of the agent, where $c(x, x') \leq B$, and for the decision maker, it includes all linear classifiers within the hypothesis class.
>
> These details will be added to the revised version of our paper, and the formally mathematical definition will be pointed out in later chapters, such as section 3.2 for the definition of the utility for the agent.
>
>
> #### **Question 2**:
>
> > ``In particular, the authors never state what the utility function for the system designer is. I think this is a critical question—the central theorem for the work should relate the outcome that is achieved by the authors' methods to the utility function of the designer. We need to know "how good" this solution is relative to others if we are going to model this game theoretically.''
>
> That's a great question. To guide our discussion, let's first define the true qualification function $y: X\rightarrow {0, 1}$ as the mapping between the feature vector $x\in R^d$ and the true qualification/label $y\in {0, 1}$.
> In the traditional strategic classification setting (see, e.g., (Hardt et al., 2016)), the system designer's payoff is $Pr_{x\sim \cal D}(h(\Delta(x)) = y(x))$, where $\Delta(x)$ is the agent's adapted feature. This means that the decision maker's utility is the classification accuracy of classifier $h$ using the adapted feature with respect to the original qualification.
> In our setting, the decision maker's goal is to classify the agents correctly with respect to their adapted feature and the corresponding new qualification.  We represent the decision maker's utility as the probability that the proposed classifier $h$ correctly predicts the label $y(\Delta(x))$ of an agent's adapted feature $\Delta(x)$. This can be expressed mathematically as $Pr(h(\Delta(x)) = y(\Delta(x)))$, where $\Delta(x)$ is the agent's adapted feature.
>
> You are absolutely right that we need to determine how good this solution is relative to others to model the game theoretically. We address this issue in Section 4, particularly in propositions 5 and 6, where we show that the terms in the proposed constructive adaptation risk defined in equation (6) provide a good approximation for the agent's utility mentioned above.
>
> In our revision, we will make sure to add the target objective function for the model designer in our paper.

---

### Review · Reviewer_5yXo · 2023-03-07

**Summary Of Contributions:**

This paper discusses an approach to designing a classifier that incentivizes strategic adaptation of improvable features (i.e. those features for which strategic adaption leads to actual improvements with respect to some target of interest) while avoiding gaming the classifying by allow adaptation with respect to manipulable features (i.e. features which are correlated with the target of interest but can be manipulated without improving the target of interest). They take as input a taxonomy of features that distinguishes improvable, manipulable and immutable features (so they are not attempting to discover this, thereby avoiding the hardness results from causal discover [Miller et al 2020]), and optimize *constructive adaptation* risk which penalizes incorrect answers as a function of the user's best response to manipulable features, and penalizes not setting the classifier to output true as a function of the best response to improvable features. That is: you want a classifier such that the best response to improvable features induces the classifier to output true, while the best response to manipulable feature just induces the classifier to output the label (so it is not more likely to classify true if the true label is false). They study this problem in the context of linear model with costs given by Mahalanobis norm of the changes in features (to account for correlations between features), which enables the analysis to benefit from closed form best responses (at the cost of generality). The results are intuitive and well presented.

**Audience:**

Yes

**Broader Impact Concerns:**

Given that this paper explicitly looks to incentivize behaviour, I would add a broader impact statement that considers what negative consequences may occur if the target, y, is poorly chosen or the assumptions are not met (e.g. accidentally including a manipulable feature may result in people who can't afford loans receiving them which is bad for both parties & if the target y is a proxy for what you want - e.g. engagement as a proxy for user happiness - I would suspect many of these notions could lead to bad outcomes?).

**Claims And Evidence:**

Yes

**Requested Changes:**

I would like to see the weaknesses outlined above addressed, but otherwise I think that the paper is very well written and ready to be accepted.

**Strengths And Weaknesses:**

Strengths:
 * The results in Section 3 give nice explicit discussion of the implications of allowing a classifier to condition on manipulable features, and the way that correlations can reduce the cost of manipulating a classifier. These results are intuitive and a little unsurprising, but given that they are very clearly presented, they serve as a useful introduction to the challenges of selecting good features in this setting.
 * Nice experimental evaluation.

Weaknesses:
 * I did not understand the condition in Prop 5. How is it ever the case that "the manipulating best response is more likely to result in a positive prediction than the unconstrained best response"? If the manipulating best response only changes manipulable features, surely its a more constrained action space, and hence it should always be less likely to result in a positive prediction (assuming the users are optimizing to induce a positive prediction)? Why can't the unconstrained best response always play the manipulating best response if that option is available?
 * I would expect the covariate shift assumption to only hold for improving features and not for manipulating features - so it seems strange to be totally black-box about the way the assumption is stated in Proposition 6. Is there any example of a manipulating feature for which the covariate shift holds? Assuming not - while you may not want to specify a specific DAG, you do have an equivalence class of DAGs that imply your assumption (I think it is the DAGs for which conditional ignorability holds given the improvable features and a subset of the immutable features), and the proposition would be improved by making that explicit.
 * I noticed that you used account balances as improvable features for the `german` data. Surely those are manipulate using the examples you give in the text? (I.e. borrowing money from a relative before the loan)

Minor:
 * Missing table reference on page 10

---

> ### Author Response · Authors · 2023-03-19
> **Response to Reviewer 5yXo**
>
> Thank you so much for your valuable feedback! Below we address your comments and requested changes:
>
>
> > "I did not understand the condition in Prop 5. How is it ever the case that "the manipulating best response is more likely to result in a positive prediction than the unconstrained best response"? If the manipulating best response only changes manipulable features, surely its a more constrained action space, and hence it should always be less likely to result in a positive prediction (assuming the users are optimizing to induce a positive prediction)? Why can't the unconstrained best response always play the manipulating best response if that option is available?"
>
> This is a great question. While it is true that the unconstrained best response generally dominates the manipulating best response, our Proposition 5 assumes that the true label of the agent remains unchanged. This means that if $y(\Delta(x) = y(x))$, then the manipulating best response is more likely to result in a positive prediction than the unconstrained best response. Intuitively, if the agent moves in the direction of manipulation rather than improvement (while keeping the true label constant), then using the manipulating best response is more efficient than using the unconstrained best response. We hope this explanation clears up any confusion.
>
> > "I would expect the covariate shift assumption to only hold for improving features and not for manipulating features - so it seems strange to be totally black-box about the way the assumption is stated in Proposition 6. Is there any example of a manipulating feature for which the covariate shift holds? Assuming not - while you may not want to specify a specific DAG, you do have an equivalence class of DAGs that imply your assumption (I think it is the DAGs for which conditional ignorability holds given the improvable features and a subset of the immutable features), and the proposition would be improved by making that explicit."
>
> Thank you for bringing this up -- we apologize for any confusion caused. To be more precise, our assumption in proposition 6 is that the distribution of the true variable Y given a particular feature vector X is unchanged. This means that the conditional probability of Y given X, namely $Pr(Y=y|X=x)$ remains the same before and after strategic manipulation. While the formula for $Pr(Y|X)$ may look like that of covariate shift, it is the probability over the random variable Y, rather than the entire distribution. Thus it is a mistake from our side to claim that our assumption is equivalent to covariate shift. We have included this clarification in our revision, specifically in proposition 6.
>
>
> > "I noticed that you used account balances as improvable features for the german data. Surely those are manipulate using the examples you give in the text? (I.e. borrowing money from a relative before the loan)"
>
> This is a great point – in our experimental section, we implicitly assume that account balances reflect the agent’s true savings. We have made this assumption explicit in our revision in the appendix on page 30.
>
> > "Missing table reference on page 10"
>
> Thanks for pointing out this! We have fixed this in our revision.

---

### Author Response · Authors · 2023-03-20
**Summary of our Revision**

We would like to thank all three reviews for your valuable feedback and suggestions again! We have individually responded to your comments, and we have made our edits accordingly in the revised version of our manuscript.

Here is a summary of the major changes we made in our revision:

* Reviewer 5yXo (colored blue):
1) Clarify the assumption we make in Proposition 6, which does not correspond to the covariate shift setting.
2) Clarify the assumption we make in our experimental section on the german dataset.
3) Add a broader impact section at the end of the paper.

* Reviewer EU7Q (colored pink):
1) Add a detailed description of the Stackelberg game, including the two parties, action spaces as well as pointers to more detailed discussions for their corresponding utility function in Section 2.1.
2) Add descriptions of the ideal utility function for the decision-maker in Section 2.3.
3) Fix the reference formatting problem.
4) Clarify the ambiguity for true qualification function $y(x)$.
5) Add a limitation on assuming the separation of the causal and non-causal features.

* Reviewer TLHt (colored red):
1) Add the GitHub repository link for the code.
2) Add the proof for Proposition 3.
3) Add a clarification on the subscript and superscript notation.



Looking forward to hearing back from you if you have more questions or feedback!

---

> ### Comment · Reviewer_EU7Q · 2023-03-20
> **Anonymity**
>
> Please remove the non-anonymous Github link from the paper.

---

> > ### Author Response · Authors · 2023-03-20
> > **Re: Anonymity**
> >
> > We have removed the github link. Thank you!

---

### Decision · Action_Editors · 2023-04-29

**Recommendation:** Accept as is

**Comment:**

The paper formalizes the problem that arises when subjects modify their features to influence their classification.  While previous work focuses on the manipulation of features that may degrade classification accuracy, this work focuses on incentivizing subjects to modify their features in a way that aligns with classification accuracy.  This is a research direction that has not received a lot of attention.  The paper provides a formalization in terms of a two-player Stackelberg game and describes suitable algorithms.  This represents an important contribution to the advancement of machine learning.  While the reviewers raised several concerns in their reviews, those concerns were addressed in the author response with suitable changes to the manuscript.  In the end, the reviewers unanimously recommend acceptance.

**Audience:**

This work will be of interest to the Machine Learning community.  More precisely, anyone in Machine Learning who may be interested in incentivizing subjects to improve their classification while preventing non-constructive manipulation should read this paper.

**Claims And Evidence:**

The proposed algorithm is supported by a theoretical analysis and an empirical evaluation that are convincing.